# KERNEL-BASED ATTRIBUTE IDENTIFICATION FOR GENERALIZABLE CLASSIFICATION

## ABSTRACT

To achieve satisfactory generalization performance on previously unseen domains, existing domain generalization (DG) methods often assume fixed domain-invariant features from a set of training domains for good generalization on new domains. However, this assumption can be overly strict, especially when the source domains lack shared information or when the target domains utilize information from selective source domains in a compositional manner. This leads to the natural question of how we utilize information from the source domain to the target domain in an appropriate way. In response to this challenge, we propose an innovative framework that includes an attribute-based feature extractor that captures from the source domains semantically meaningful components referred to as *attributes* and a *Kernel-based Attribute Identifier* that leverages kernel learning theory to define the decision boundaries for these attributes collected from the source domains. This dynamic learning approach empowers the classifier to effectively identify the learned attributes in the domains it has not encountered before. We empirically validate our method on well-established DG benchmarks, achieving competitive results compared to state-of-the-art techniques.

## 1 INTRODUCTION

One of the most challenging problems in applying machine learning to real-world problems is to address the domain shift encountered when test data at inference time come from different distributions compared to training data, often causing unexpectedly imperfect generalization performance. To handle this issue, many out-of-distribution learning settings have been investigated, notably domain adaptation (DA) and domain generalization (DG). In particular, DA setting (Mansour et al., 2009; Ben-David et al., 2010; Zhao et al., 2019; Phung et al., 2021) takes the assumption that both labeled source data and unlabeled target data are available at the training phase, while DG setting (Blanchard et al., 2011; Muandet et al., 2013; Ganin et al., 2016) is much more challenging due to the complete absence of any target data at training time. Furthermore, the learned models are expected to perform a zero-shot prediction on test samples. While being more challenging than DA, DG is arguably more versatile and applicable to real-world scenarios where there is a need to rapidly deploy a prediction model on a new target domain without any access to target data.

Many existing approaches aimed at addressing the domain shift problem rely on shared features and learning concepts across different source domains (Muandet et al., 2013; Ganin et al., 2016; Motiian et al., 2017; Ghifary et al., 2015; Xie et al., 2017; Wang et al., 2019; Piratla et al., 2020; Zhao et al., 2020; Nguyen et al., 2021). These features, often referred to as domain-invariant representations, are trained to capture a latent representation from multiple source domains that can generalize to unseen target domains, thus mitigating domain shift issues. These domain invariance approaches have limitations, primarily because the latent space is typically high-dimensional, and unseen target domains may exhibit significant and unexpected variations. In real-world data, labels often depend on multiple attributes, and different groups of domains may share distinct sets of these attributes. In this context, the target domain's similarity to one or more source domains is a complex interplay. Enforcing domain-invariant learning across all source domains may lead to the exclusion of valuable attributes during test time.

Recent research (Huang et al., 2020; Bui et al., 2021; Chattopadhyay et al., 2020) has underscored the advantages of incorporating domain-specific knowledge for enhanced generalization. A critical

challenge encountered in this context is the identification of pertinent attributes unique to the target domain. This often leads to the utilization of all learned attributes for all target domains. *We argue that such behavior is sub-optimal. Specifically, when encountering a new example from an unfamiliar target domain, which comprises various attribute sets, the classifier needs to be informed about the attributes it has learned from the source domain to accurately predict the correct label.*

Our goal in this work, therefore, is to propose a mechanism to alert the classifier of the presence of previously acquired attributes in the target domain when making predictions. To accomplish this, we propose a novel **KE**rnel-based **A**ttribute **I**dentification (KEAI) framework including: (i) attribute-based feature extractor, which aim to capture meaningful components in source domains (See formal definition 1), and (ii) *Kernel-based Attribute Identifier*. This identifier uses kernel learning theory to define and refine the decision boundaries of *attributes* from source domains, enabling our model to identify familiar attributes even in new domains. Finally, we empirically demonstrate that our proposed method can achieve favorable results when evaluating our model on domain generalization benchmarks in the comparison with state-of-the-art methods.

## 2 PRELIMINARIES

In this section, we provide essential background information, including the intuition behind how kernel methods contribute to clustering and out-of-distribution (OOD) detection, as well as an introduction of random Fourier features, which play a crucial role in integrating kernel methods into our proposed framework.

### 2.1 CLUSTERING INDUCED AND OOD DETECTION VIA KERNEL LEARNING

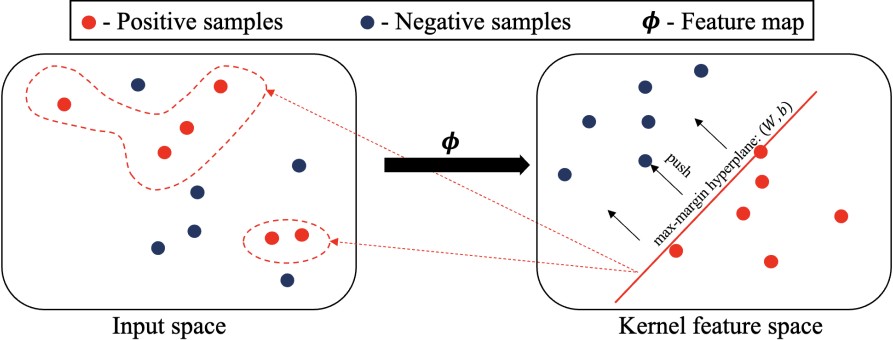

Figure 1: Illustration of clustering induced via kernel learning: the max-margin hyperplane, when mapped back to the latent space forms sets of contours tightly enclosing corresponding (positive side) data points into clusters.

The kernel method is widely used in support vector clustering and clustering analysis (Ben-Hur et al., 2001; Nguyen et al., 2018), and data description (Tax & Duin, 2004) and one-class support vector machine for anomaly detection (Schölkopf et al., 1999). In these techniques, data points are transformed from the original data space to a higher-dimensional feature space using a kernel e.g., Gaussian kernel. In the kernel feature space, the objective is to identify the smallest sphere that encompasses the transformed data points or the max-margin hyperplane that separates the transformed data of interest from the negative examples (Schölkopf et al., 1999; Tax & Duin, 2004). The learned sphere or hyperplane is then projected back into the original input space, resulting in a series of contours that envelop the target points. These contours serve as cluster boundaries, with points within each distinct contour being assigned to the same cluster. Additionally, it is found that these contours effectively outline the support of the underlying probability distribution, essentially highlighting high-density regions in the distribution's landscape. This characteristic is particularly valuable for outlier detection. For example in Figure.1, by introducing negative examples (represented by blue points) alongside target samples (red points), we can learn the hyperplane in the kernel feature space separates kernel features of positive and negative samples on kernel feature space, resulting to a

series of contours that encompass the positive points on input space. These cluster boundaries can be utilized to identify novel data points or outliers.

## 2.2 REPARAMETERIZED RANDOM FOURIER FEATURE

Let $x \in \mathbb{R}^N$ denote the $N$-dimensional vector in data domain $\mathcal{X}$. The vanilla kernel methods define an implicit lifting $\phi(x)$ from data space to feature space, and the inner product $\langle \phi(x), \phi(x') \rangle$ is evaluated through a positive semi-definite kernel $\kappa(x, x')$ using the so-called kernel trick. To construct an explicit representation of $\phi(x)$, the key idea is to approximate the original kernel $\kappa(x, x')$ using a kernel induced by a random finite-dimensional feature map (Rahimi & Recht, 2007; Nguyen et al., 2017), i.e., given $e_d \overset{i.i.d}{\sim} \mathcal{N}(\omega \mid 0, I)$, we can construct a random feature map $\hat{\phi}_{\sigma, D} : \mathcal{X} \to \mathbb{R}^{2D}$, termed reparameterized random feature (RRF), wherein $\tilde{\phi}_{\sigma, D}^\top$ is given as:

$$\tilde{\phi}_{\sigma, D}^\top(x) = \left[ \cos\left( (\mathrm{diag}\,(\sigma)\, e_d)^\top x \right), \sin\left( (\mathrm{diag}\,(\sigma)\, e_d)^\top x \right) \right]_{d=1}^D / \sqrt{D} \tag{1}$$

resulting in the induced kernel $\tilde{\kappa}(x, x')_\omega = \tilde{\phi}_{\sigma, D}(x)^\top \tilde{\phi}_{\sigma, D}(x)$ that can accurately and efficiently approximate the original kernel: $\tilde{\kappa}(x, x')_\omega \approx \kappa_\sigma(x, x')$.

## 3 PROPOSED METHOD

### 3.1 PROBLEM SETTING

We consider *Domain-free* DG setting, where there is no domain information during training. Therefore, we simply take their union without using the original domain labels as the source domain, i.e., $\mathbb{D}^S = \{(x^i, y^i)\}_{i=1}^{N^S}$ be the source labeled datasets; $N^S$ denotes the number of examples in $\mathbb{D}^S$, and $y_i \in \mathcal{Y} := \{1, ..., C\}$ represents the instance ground truth with a set of $C$ classes. The goal of DG is to learn a model $f$ on source domains $\mathbb{D}^S$ that is expected to perform well on target domains $\mathbb{D}^T = \{(x^{T,i}, y^{T,i})\}_{i=1}^{N^T}$.

In general, we examine the composite hypothesis $f = h \circ g$, where $g : \mathcal{X} \to \mathcal{Z}$ is an encoder mapping the data space to a latent space and $h : \mathcal{Z} \to \mathcal{Y}$ is the classifier on this latent space. Let $\ell_{cls}(h(g(x)), y)$ be the loss incurred by using this hypothesis to predict the label of $x \in \mathcal{X}$, given its ground-truth one $y \in \mathcal{Y}$. The general loss of the hypothesis $f$ w.r.t. the joint distribution $\mathbb{D}$ is:

$$\mathcal{L}(f, \mathbb{D}) = \mathbb{E}_{(x,y) \sim \mathbb{D}} \left[ \ell_{cls}(h(g(x)), y) \right]. \tag{2}$$

### 3.2 OVERALL PROPOSED FRAMEWORK

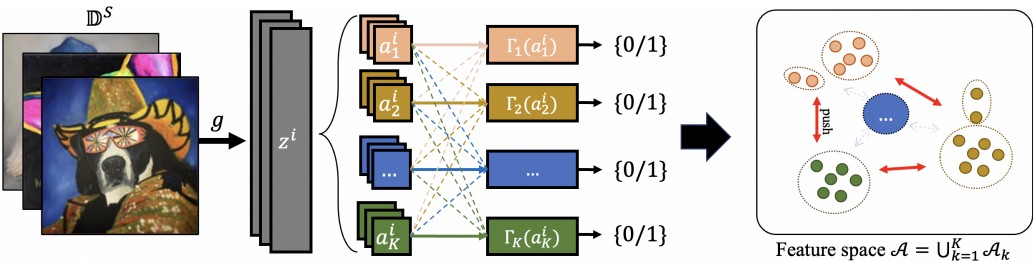

Figure 2: The overall Kernel-based Attribute-Identification for DG Framework, in which, *attributes* of each basis are characterized into clusters by sets of contours. Additionally, clusters of different bases mutually push others far away, allowing the model to avoid attributes-collapse on feature space.

Our approach centers on acquiring knowledge about semantically meaningful factors within source domains and discerning their presence within the target domain to make prediction. The overall framework is depicted in Figure 2. Drawing inspiration from the concept of disentangled representation learning, given an encoder $g$, we break down the learned features $z^i = g(x^i)$ into $K$ sub-components, which refer to as *attributes*. The set of all $k$-th attributes $\mathcal{A}_k = \{a_k^i\}_{i=1}^{N^S}$ is called a basis, i.e., $k^{th}$

basis (See formal definition 1). Subsequently, we leverage kernel learning theory to characterize the attributes of each basis by contours (max-margin hyperplane $\Gamma_k$ on kernel feature space) which can be interpreted as separated clusters tightly enclosing attributes in the attribute space (Figure 2.Right). This capability empowers our model to proficiently identify learned attributes from source domains, even in domains it has not previously encountered.

### 3.3 ATTRIBUTE-BASED REPRESENTATION

We begin by describing the construction of attribute-based Representation. Particularly, we initially extract a latent vector for each data point $x^i$ using an encoder $z^i = g(x^i) \in \mathbb{R}^{n_z}$. We then partition the latent vector z into K groups, each with equal dimensionality $n_a$ ($n_z = K \times n_a$), denoted as $z^i = [a_1^i, a_2^i, ..., a_K^i]^\top$ where each $a_k^i = g(x_i)_k \in \mathbb{R}^{n_a}$ is denoted as *attribute* representing semantically meaningful concepts. Consequently, given the mixture of source dataset $\mathbb{D}^S$ and encoder $g$, we have:

$$
\left\{ z^1, z^2, ..., z^N \right\} = \left\{ \begin{bmatrix} a_1^1 \\ a_2^1 \\ ... \\ \underline{a_k^1} \\ ... \\ a_K^1 \end{bmatrix}, \begin{bmatrix} a_1^2 \\ a_2^2 \\ ... \\ \underline{a_k^2} \\ ... \\ a_K^2 \end{bmatrix}, \quad ... \quad , \begin{bmatrix} a_1^{N^S} \\ a_2^{N^S} \\ ... \\ \underline{a_k^{N^S}} \\ ... \\ a_K^{N^S} \end{bmatrix} \right\}
$$

Then, we define $\mathcal{A}_k = \left\{ a_k^1, a_k^2, ..., a_k^{N^S} \right\}$ as $k$-th basis which contains $k$-th attribute of all samples.

**Definition 1. (Basis of attributes)** For a dataset $\mathbb{D} = \{(x^i, y^i)\}_{i=1}^N$ and feature extractor $g$, a set $\mathcal{A}_k = \left\{ a_k^1, a_k^2, ..., a_k^N \right\}$ where each $a_k^i = g(x^i)_k$ is considered as *basis of attributes* with respect to $(\mathbb{D}, g)$ if there exists a function $h_k$ such that $h_k(a_k^i) = y^i, \forall i = 1...N$.

Definition 1 states that $\mathcal{A}_k$ is considered as a basis of attributes that encapsulates meaningful concepts within $\mathbb{D}^S$ if any element $a_k^i \in \mathcal{A}_k$ can be used to determine its corresponding class-label $y^i$. To achieve this, we learn an encoder $g$ and a set of attribute-based classifiers $h = \{h_k : \mathcal{A} \to \mathcal{Y}\}_{k=1}^K$ by optimizing the following objective function:

$$
\mathcal{L}_{\text{Attribute}}^k = \frac{1}{N^s} \sum_{i=1}^{N^S} \ell_{cls} \left( h_k \left( g \left( x^i \right)_k \right), y^i \right), \forall k = 1...K \tag{3}
$$

Next, we introduce a set *Kernel-based Attribute Identifier* $\Gamma = \{\Gamma_k\}_{k=1}^K$, where each $\Gamma_k$ is responsible for *characterizing* and *refining* the *attributes* in the basis $\mathcal{A}_k$ in the sense that $\Gamma_k$ is able to proficiently identify whether a new attribute belongs to the basis $\mathcal{A}_k$ or not. To elaborate, considering the output of $\Gamma_k$ falls in $\{0, 1\}$ i.e., given $a_k$ is the $k$-th attribute of a new datapoint, $\Gamma_k(a_k) = 1$ signifies that the attribute $a_k$ belongs to the basis $\mathcal{A}_k$, while $\Gamma_k(a_k) = 0$ indicates otherwise. For a detailed explanation of how we model $\Gamma$, please refer to the following section.

Our focus is solely on attributes present in the source domains, as the attribute-based classifiers $\{h_k\}_{k=1}^K$ are trained to predict based on the observed attributes in the source domains. Along with attribute-based classifiers, this leads to the prediction $f(x) = \frac{\sum_{k=1}^K \Gamma_k(g(x)_k) h_k(g(x)_k)}{\sum_{k=1}^K \Gamma_k(g(x)_k)}$ which is considered as the ensemble prediction of selected attributes.

### 3.4 KERNEL-BASED ATTRIBUTE IDENTIFIER

In this section, we describe how the kernel method is employed to represent observed attributes within each basis $\mathcal{A}_k$. More precisely, each *Kernel-based Attribute Identifier* $\Gamma_k$ is modelled as a max-margin hyperplane $(W_k, b_k)$, having equation $W_k^\top \phi_\sigma(a) + b_k = 0$ where attribute $a$ is the input and $\phi_\sigma$ is the mapping from attribute space to kernel space. To be able to integrate kernel models nicely to deep nets, we utilize random Fourier features (Rahimi & Recht, 2007; Nguyen et al., 2017) to approximate an RBF kernel for $\phi_\sigma$. Particularly, the feature map $\phi_\sigma : \mathcal{U} \to \mathbb{R}^{2 \times D}$ admits the following form:

$$\phi_\sigma^\top(x) = \left[\cos\left(\left(\operatorname{diag}(\sigma)\, e_d\right)^\top x\right), \sin\left(\left(\operatorname{diag}(\sigma)\, e_d\right)^\top x\right)\right]_{d=1}^D / \sqrt{D} \tag{4}$$

where $e_d \overset{i.i.d}{\sim} \mathcal{N}(0, I)$; $2 \times D$ is the dimension of kernel feature space and $\sigma$ is kernel width.

Let's recall that the objective of $\Gamma_k$ is to characterize attributes within the basis $\mathcal{A}_k$. Our approach involves utilizing attributes from other bases as negative samples. Specifically, the max-margin hyperplane $\Gamma_k$ is built in such way that (i) it separates push-forward kernel-features of positive attributes $C_{+k} = \{\phi_\sigma(a) \mid a \in \mathcal{A}_k\}$ and the negative attributes $C_{-k} = \{\phi_\sigma(a) \mid a \in \mathcal{A}_{i \neq k}\}$ and (ii) the margin w.r.t the hyperplane $\Gamma_k$ defined as the closest distance from negative in $C_{-k}$ to this hyperplane is maximized. This construction method results in a hyperplane that can be aptly interpreted as cluster-boundaries that tightly envelop attributes belonging to the basic $\mathcal{A}_k$. It's noteworthy that our max-margin hyperplane differs from the example illustrated in Figure 1, primarily because our hyperplane is integrated with the encoder $g$, which allows it to refine attribute clusters from basic $\mathcal{A}_k$ in a more compact manner while effectively separating them from attributes originating in other bases. Consequently, this leads us to the following optimization problem (OP):

$$(W_k, b_k) = \arg\max_{W_k, b_k} \min_{a \in C_{-k}} \frac{\left|W_k^\top \phi_\sigma(a) + b_k\right|}{\|W_k\|} \tag{5}$$
$$\text{s.t: } W_k^\top \phi_\sigma(a) + b_k \geq 0 \text{ for } a \in C_{+k}$$
$$W_k^\top \phi_\sigma(a) + b_k \leq 0 \text{ for } a \in C_{-k}$$

Since the margin is invariant if we replace $(W_k, b_k)$ by $(\lambda W_k, \lambda b_k)$ for any $\lambda > 0$, without the loss of generalization, we can naturally assume that $\min_{u \in C_{-k}} \left|W_k^\top \phi_\sigma(u) + b_k\right| = 1$, hence rewriting the above OP as:

$$(W_k, b_k) = \arg\max_{W_k, b_k} \frac{1}{\|W_k\|} \text{ or } \arg\min_{W_k, b_k} \|W_k\|$$
$$\text{s.t: } W_k^\top \phi_\sigma(u) + b_k \geq 0 \text{ for } u \in C_k$$
$$W_k^\top \phi_\sigma(u) + b_k \leq -1 \text{ for } u \in C_{-k}$$

Using the slack variables $\xi$, we develop the soft version as

$$(W_k, b_k) = \arg\min_{W_k, b_k} \|W_k\|$$
$$\text{s.t: } W_k^\top \phi_\sigma(u) + b_k \geq -\xi_k \text{ for } u \in C_k$$
$$W_k^\top \phi_\sigma(u) + b_k \leq -1 + \xi_k \text{ for } u \in C_{-k}$$

Note that for the optimal solution $\xi_{k,a} = \max\{0, -W_k^\top \phi_\sigma(a) - b_k\}$ for $u \in C_{+k}$ and $\xi_{k,a} = \max\{0, -1 + W_k^\top \phi_\sigma(a) + b_k\}$ for $a \in C_{-k}$, we arrive at the following OP:

$$\mathcal{L}_{\text{Kernel}}^k = \frac{\lambda}{2}\|W_k\|_2^2 + \frac{1}{|C_k|}\sum_{a \in C_{+k}} \max\{0, -W_k^\top \phi_\sigma(a) - b_k\}$$
$$+ \frac{1}{|C_{-k}|}\sum_{a \in C_{-k}} \max\{0, -1 + W_k^\top \phi_\sigma(a) + b_k\} \tag{6}$$

where $|\cdot|$ represents the cardinality of a set and $\lambda > 0$ is a trade-off parameter.

Finally, for new attribute $a$, the indicator $\Gamma_k(a) = \mathbb{1}_{W_k^\top \phi_\sigma(a) + b_k \geq -\xi_k}$ indicates that $a$ is inside tight clusters of observed attributes of $k$-th basis or not. In simpler terms, $\Gamma_k(a)$ serves as a decision-maker, indicating whether attribute $a$ is an observed attribute from the source domains or not.

**Remark:** Remind that the of the max-margin hyperplane $\Gamma_k$ when mapped back into the feature space $\mathcal{A}_k$ becomes the set of contours tightly enclosed positive attributes into clusters, excluding negative attributes (Ben-Hur et al., 2001; Tax & Duin, 2004; Schölkopf et al., 1999). Therefore, simultaneously optimizing $\Gamma = \{\Gamma_k\}_{k=1}^K$ in join space $\mathcal{A} = \cup \mathcal{A}_k$ will enforce clusters of *attributes* from different basis separate to each other (as illustrated in 2.Right), allowing model capture diverse attributes.

## 3.5 Overall Framework

**Training:** Combining the *attribute-based function* loss and *Kernel-based Attribute Identifier* loss functions, we propose to jointly minimize:

$$\min_{g,\{h_k\}_{b=1}^K,\{\Gamma_k\}_{b=1}^K} \sum_{k=1}^K \left( \mathcal{L}_{\text{Attribute}}^k + \alpha \mathcal{L}_{\text{Kernel}}^k \right) \tag{7}$$

where $\alpha > 0$ is the trade-off hyper-parameter.

**Inference:** For a new sample, we utilize $\Gamma$ to identify attributes which are observed in source domains to make predictions: i.e., $\hat{y} = f(x) = \frac{\sum_{k=1}^K \Gamma_k(g(x)_k) h_k(g(x)_k)}{\sum_{k=1}^K \Gamma_k(g(x)_k)}$ .

Finally, the pseudo-code of our KEAI is summarized in Algorithm 1.

---

**Algorithm 1** KEAI

---

1: Initialize: encoder $g$, classifier $h$, *Sub-space indicator* $\Gamma$ and dataset $\mathbb{D}^S$.
2: **for** epoch $= 1 \to$ epochs **do**
3:     **for** iter **in** iterations **do**
4:         Sample Mini-batch: $\mathbb{B} = \left\{ \left( x^1, y^1 \right), \left( x^2, y^2 \right), ..., \left( x^B, y^B \right) \right\} \sim \mathbb{D}^S$
5:         Optimize $\mathcal{L}_{\text{Kernel}}$ w.r.t. $\Gamma$ on $\mathbb{B}$.
6:         Optimize $\mathcal{L}_{\text{Attribute}}$ w.r.t. $h$, $g$ on $\mathbb{B}$.
7:     **end for**
8: **end for**
9: **Return:** The optimal: $g^*$, $h^*$ and $\Gamma^*$.

---

## 4 Experiment

### 4.1 Main Results on Benchmark Datasets for Multi-source DG

Table 1: Classification accuracy (%) for all algorithms and datasets summarization. The **best** and second best results are highlighted in hold and underline.

| Algorithm | VLCS | PACS | OfficeHome | TerraIncognita | DomainNet | Avg |
|---|---|---|---|---|---|---|
| ERM (Gulrajani & Lopez-Paz, 2021) | $77.5 \pm 0.4$ | $85.5 \pm 0.2$ | $66.5 \pm 0.3$ | $46.1 \pm 1.8$ | $40.9 \pm 0.1$ | 63.3 |
| IRM (Arjovsky et al., 2019) | $78.5 \pm 0.5$ | $83.5 \pm 0.8$ | $64.3 \pm 2.2$ | $47.6 \pm 0.8$ | $33.9 \pm 2.8$ | 61.6 |
| GroupDRO (Sagawa et al., 2019) | $76.7 \pm 0.6$ | $84.4 \pm 0.8$ | $66.0 \pm 0.7$ | $43.2 \pm 1.1$ | $33.3 \pm 0.2$ | 60.7 |
| Mixup (Wang et al., 2020) | $77.4 \pm 0.6$ | $84.6 \pm 0.6$ | $68.1 \pm 0.3$ | $47.9 \pm 0.8$ | $39.2 \pm 0.1$ | 63.4 |
| MLDG (Li et al., 2018a) | $77.2 \pm 0.4$ | $84.9 \pm 1.0$ | $66.8 \pm 0.6$ | $47.7 \pm 0.9$ | $41.2 \pm 0.1$ | 63.6 |
| CORAL (Sun & Saenko, 2016) | $78.8 \pm 0.6$ | $86.2 \pm 0.3$ | **$68.7 \pm 0.3$** | $47.6 \pm 1.0$ | $41.5 \pm 0.1$ | 64.5 |
| MMD (Li et al., 2018b) | $77.5 \pm 0.9$ | $84.6 \pm 0.5$ | $66.3 \pm 0.1$ | $42.2 \pm 1.6$ | $23.4 \pm 9.5$ | 58.8 |
| DANN (Ganin et al., 2016) | $78.6 \pm 0.4$ | $83.6 \pm 0.4$ | $65.9 \pm 0.6$ | $46.7 \pm 0.5$ | $38.3 \pm 0.1$ | 62.6 |
| CDANN (Li et al., 2018b) | $77.5 \pm 0.1$ | $82.6 \pm 0.9$ | $65.8 \pm 1.3$ | $45.8 \pm 1.6$ | $38.3 \pm 0.3$ | 62.0 |
| MTL (Blanchard et al., 2021) | $77.2 \pm 0.4$ | $84.6 \pm 0.5$ | $66.4 \pm 0.5$ | $45.6 \pm 1.2$ | $40.6 \pm 0.1$ | 62.9 |
| SagNet (Nam et al., 2021) | $77.8 \pm 0.5$ | $86.3 \pm 0.2$ | $68.1 \pm 0.1$ | **$48.6 \pm 1.0$** | $40.3 \pm 0.1$ | 64.2 |
| ARM (Zhang et al., 2021) | $77.6 \pm 0.3$ | $85.1 \pm 0.4$ | $64.8 \pm 0.3$ | $45.5 \pm 0.3$ | $35.5 \pm 0.2$ | 61.7 |
| VREx (Krueger et al., 2021) | $78.3 \pm 0.2$ | $84.9 \pm 0.6$ | $66.4 \pm 0.6$ | $46.4 \pm 0.6$ | $33.6 \pm 2.9$ | 61.9 |
| RSC (Huang et al., 2020) | $77.1 \pm 0.5$ | $85.2 \pm 0.9$ | $65.5 \pm 0.9$ | $46.6 \pm 1.0$ | $38.9 \pm 0.5$ | 62.7 |
| KEAI | **$79.4 \pm 0.3$** | **$86.8 \pm 0.1$** | $68.4 \pm 0.1$ | $48.6 \pm 1.3$ | **$41.8 \pm 0.3$** | **65.0** |

Table 1 reports the results of our experiments on 5 benchmark datasets when compared with mentioned methods. The full result per dataset and domain is provided in Appendix B. Our model achieves comparable or better performances on most datasets and obtains average 0.3 points improvement on all datasets. Those empirical results clearly indicate that our KEAI provides competitive classification accuracy compared to the baselines.

### 4.2 Single-source DG

To further demonstrate the effectiveness of our method, we evaluate KEAI on a more challenging DG task with a single source domain. It is worth noting that *Photo* is considered to contain rich

information to predict the label in comparison with three other domains. Therefore, we conduct experiments on PACS, where Photo is chosen as the source domain, and the remaining domains (i.e., Art painting, Cartoon, and Sketch) are selected as the target domains. As shown in Table 2, the experimental results consistently indicate that KEAI achieves superior performance in terms of test accuracy on all the target domains.

Table 2: Single-source DG on PACS with "Photo" is selected as the source domain for training.

| Method | Art-painting | Cartoon | Sketch | Ave |
|---|---|---|---|---|
| ERM (Gulrajani & Lopez-Paz, 2021) | $60.7 \pm 0.00$ | $23.5 \pm 0.00$ | $29.0 \pm 0.00$ | 37.70 |
| JiGen (Carlucci et al., 2019) | $63.6 \pm 0.00$ | $28.5 \pm 0.00$ | $30.2 \pm 0.00$ | 40.80 |
| CrossGra (Shankar et al., 2018) | $64.2 \pm 0.00$ | $29.4 \pm 0.00$ | $32.1 \pm 0.00$ | 41.90 |
| DDAIG (Zhou et al., 2020b) | $64.1 \pm 0.00$ | $\underline{32.5} \pm 0.00$ | $29.6 \pm 0.00$ | 42.10 |
| M-ADA (Qiao et al., 2020) | $64.6 \pm 0.00$ | $34.6 \pm 0.00$ | $26.6 \pm 0.00$ | 41.90 |
| KEAI | $\mathbf{73.6} \pm 0.07$ | $\mathbf{38.6} \pm 1.72$ | $\mathbf{41.3} \pm 3.18$ | **51.18** |

## 4.3 ABLATION EXPERIMENTS

### 4.3.1 ATTRIBUTE-BASED REPRESENTATION VISUALIZATION

As the kernel-models (hyperplanes) $\{\Gamma_k\}_1^K$ characterize high-density regions in the distribution of target attributes, our model identify whether an attribute in the target domain was learned in the source domain based on set of kernel-model $\{\Gamma_k\}_1^K$ in the sense that new attribute in target domain is lied on high-density regions in the distribution of learned attributes from source domain or not. However, we lack ground-truth labels for qualitative assessment. To gain a deeper understanding of the advantages offered by the "Kernel-based Attribute Identifier" (KEAI), we employ t-SNE (van der Maaten & Hinton, 2008) to visualize the distribution of learned attributes.

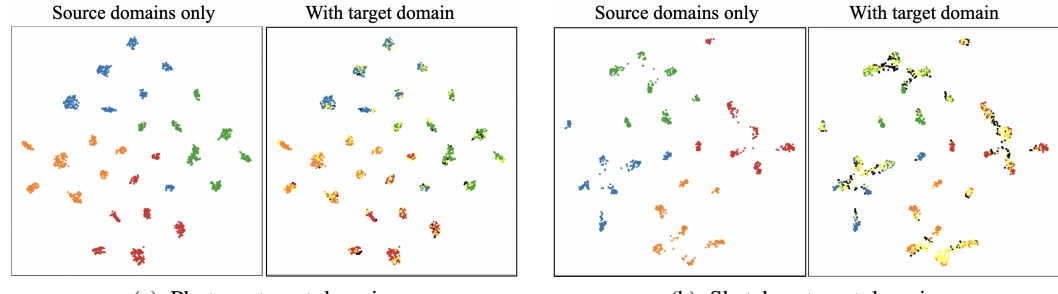

(a) Photo as target domain.          (b) Sketch as target domain..

Figure 3: The t-SNE feature visualization of attributes from the first four basics learned from KEAI. For source domains only figures, different colors (**red**, **orange**, **green** and **blue**) represent different basic-labels. Target domains are presented in **black** for deselected attributes and **yellow** for selected attributes.

Figure 3 provides illustrations of the attribute space generated by KEAI. For our analysis, we select the *Photo* which contains rich information to predict the label in comparison with three other domains as the target domain, and *sketch* domain, which exclusively comprises colorless images, as the target domain. This domain is intentionally chosen because it is the most distant from the others, resulting in the largest source-target divergence. Additionally, given the complexity of visualization, we focus on plotting the attribute distributions of the first four basics in a single figure.

**Attribute Identification.** Notably, the visualization from Figure 3a.Left and Figure 3b. Left demonstrates that attributes from different basics are distinctly clustered, indicating effective separation. This observation demonstrates the benefits of *Kernel-based Attribute Identifier* (1) facilitating the identification of attributes, and (2) enhancing the diversity of attributes. Indeed, it can be seen from Figure 3a. Right and Figure 3b. Right, the selected attributes of the target domain are situated within the clusters of attributes from the source domains while deselected attributes (3b.Right) are located

outside them. This observation underscores the effectiveness of the *Kernel-based Attribute Identifier* in identifying observed attributes in target domains.

Additionally, we observe that the *photo* domain effectively utilizes attributes from all three source domains, as most of its attributes are selected (Figure 3a.Right). In contrast, the *sketch* domain (Figure 3b.Right) shows fewer selected attributes.

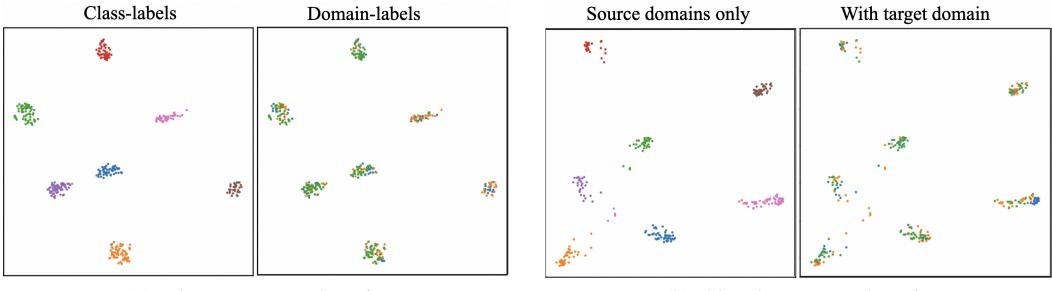

(a) Photo as target domain.                    (b) Sketch as target domain.

Figure 4: The t-SNE feature visualization of attributes from first basics learn from KEAI in domain-label and class-label levels.

**Attribute Representation.** it's important to observe that some clusters share the same color in the initial two figures. To gain a deeper understanding of this phenomenon, we further visualize the first basic at both the class-label level (Figure 4a) and the domain-label level (Figure 4b). This visualization highlights that attributes with the same class-label are distinctly distributed into separate clusters when examined from a class-label perspective. However, when considered from a domain-label perspective, these attributes tend to intermingle within clusters. This observation underscores the expressive power of the kernel-based attribute representation, as attributes are meaningfully distributed with respect to class-label, aligning with our definition of *basis of attributes*.

### 4.3.2 SENSITIVITY OF FEATURE-SIZE (FS)

Table 3: Classification Accuracy on PACS using ResNet50 with different feature-size ($n_u$).

| $\mathbb{R}^{n_a}$ | $K$ | Photo | Art-painting | Cartoon | Sketch | Ave |
|---|---|---|---|---|---|---|
| 128 | 16 | **97.8** ± 0.59 | 86.0 ± 0.31 | 77.7 ± 0.81 | 75.3 ± 0.85 | 84.19 |
| 64 | 32 | 97.6 ± 0.12 | **88.5** ± 0.42 | 78.8 ± 0.48 | 76.4 ± 0.53 | 85.32 |
| 32 | 64 | 97.5 ± 0.04 | 87.4 ± 0.17 | **82.0** ± 0.91 | **80.2** ± 0.38 | **86.76** |
| 16 | 128 | 97.7 ± 0.34 | 87.4 ± 0.65 | 78.4 ± 0.29 | 78.8 ± 0.18 | 85.56 |
| ERM | 1 | 97.2 ± 0.30 | 84.7 ± 0.40 | 80.8 ± 0.60 | 79.3 ± 1.00 | 85.50 |

We also conduct the ablation study on the sensitivity of FS to kernel-based feature learning. Specifically, we examined different feature sizes, namely $16, 32, 64,$, and $128$. The results, presented in Table 3, demonstrate that KEAI consistently improves the performance of "Photo" and "Art-painting" across various feature sizes. However, the performance of "Cartoon" and "Sketch" exhibits a strong dependence on the chosen feature size. This finding indicates that the *Kernel-based Attribute Identifier* is notably sensitive to changes in feature size.

## 5 RELATED WORK

Domain generalization approaches can be grouped into domain-invariant representation learning, meta-learning, and augmentation/self-supervision. Domain-invariant representation learning aims to learn a domain-invariant representation that can transfer well to unseen domains. Notably, Muandet et al. (2013) and Xie et al. (2017) construct shared components by minimizing the discrepancy of the source domain marginal distributions using a kernel-based algorithm and an adversarial training strategy, respectively. Seo et al. (2020) combine batch normalization and instance normalization to

remove domain-specific styles while preserving semantic category information. Autoencoder-based methods have also been proposed, such as multi-view autoencoders (Ghifary et al., 2015), adversarial autoencoders combined with Maximum Mean Discrepancy (MMD) measure (Li et al., 2018b), and variational autoencoder for representation disentanglement (Ilse et al., 2020), but these methods are hard to be applied for real-world applications due to the limitation of the autoencoder. Li et al. (2022) promotes the learning of invariant representations by invariant information Bottleneck principle. Chan et al. (2022) aims to find an invariant linear discriminative representation of data by optimizing the rate reduction objective. Recently, learning domain-specific information to boost classification performance in DG has attracted more attention. Huang et al. (2020) iteratively discard dominant features to exploit all useful features. Chattopadhyay et al. (2020) propose a domain-specific mask learned from the domain label to balance domain-invariant and domain-specific features. Bui et al. (2021) explicitly disentangle domain-invariant and domain-specific features and utilize a meta-training scheme to support domain-specific information adaptation from source domains to unseen domains. Meta-learning is another efficient approach for DG. Li et al. (2018a) and Balaji et al. (2018) simulate meta-train/meta-test using source domains. Dou et al. (2019) extend Li et al.'s work with metric learning loss to encourage domain-independent semantic feature space, while Shi et al. (2022) propose an inter-domain gradient matching objective to learn invariant features. Self-supervised learning (Yao et al., 2022) and data augmentation have also been applied to DG. Carlucci et al. (2019) propose solving the pretext task of Jigsaw Puzzles to improve generalization to unseen domains, while Shankar et al. (2018) augment training data with instances perturbed along with directions of domain change. Zhou et al. (2020a) employ a classifier that can learn the generalization on additional augmented samples of diversity pseudo-novel domains by leveraging optimal transport theory. Zhou et al. (2020b) augment the training data of source domains with synthetic data from unseen domains that can fool the domain classifier to make the task model more domain-generalizable, while Zhou et al. (2021) mix the styles of different source domains based on normalization-based style-transfer technique. Xu et al. (2021) mix the styles of training instances across domains by mixing amplitude spectrums. Kang et al. (2022) create new styles from both the styles seen in the source and those previously synthesized. Kernel-based methods have seen widespread use and extensive study within the domain of DG. Often, these methods are closely linked to other categories, serving primarily as measures of divergence or similarity. For instance, Blanchard et al. (2021) utilized positive semi-definite kernel learning to develop a domain-invariant kernel from training data. Works by Grubinger et al. (2015); Pan et al. (2010); Muandet et al. (2013) have adapted domain component analysis to address discrepancies in the marginal distribution across multi-domains within the feature space. Li et al. (2018b) focused on learning a feature representation with a domain-invariant class-conditional distribution. (Ghifary et al., 2016) employed Fisher's discriminant analysis to reduce the discrepancy of representations within the same class and domain while enhancing the discrepancy across different classes and domains. Additionally, Hu et al. (2020) introduced multi-domain discriminant analysis for class-wise kernel learning.

Diverging from these existing methods, our proposed approach also falls under kernel-based methods but takes a distinct route. Unlike traditional techniques that utilize the kernel as a tool for measuring divergence or similarity and rely on the kernel trick to avoid explicit computation of the feature map, our method takes a direct approach. We compute the feature map through a random finite-dimensional feature map (Rahimi & Recht, 2007; Nguyen et al., 2017). This enables us to leverage a clustering-based perspective of kernel methods, which is instrumental in characterizing and refining representations for effective domain generalization.

## 6 CONCLUSION

In this study, we introduce a novel framework designed to alert the classifier to the presence of previously acquired attributes in the target domain during predictions. This framework, KEAI (Kernel-based Enhanced Attribute Identification), consists of two key modules: (i) attribute-based feature extractor, which aim to capture meaningful components in source domains, (ii) *Kernel-based Attribute Identifier*, leveraging kernel learning theory to delineate the decision region of *attributes* and refine attribute-representation collected from the source domains. This empowers the model to detect learned attributes in unseen domains. Our experimental results on benchmark datasets provide clear evidence that our proposed method, KEAI, performs competitively when compared to the current state-of-the-art models.

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

APPENDIX

This supplementary material is organized as follows:

- In Section 7, we present the experimental settings and detailed results of each dataset.
- In Section 8, we present the implementation specification of our proposed approach.

## 7 FULL RESULTS

**Metric.**  we adopt the training and evaluation protocol as in DomainBed benchmark (Gulrajani & Lopez-Paz, 2021), including dataset splits, hyperparameter (HP) search, model selection on the validation set, and optimizer HP. However, we use a reduced HP search space to reduce computational costs. For training, we choose one domain as the target domain and the remaining domains as the training domain, with 20% of the samples used for validation and model selection.

### 7.1 DATASETS

To evaluate the effectiveness of the proposed method, we utilize five datasets: PACS (Li et al., 2017), VLCS (Torralba & Efros, 2011), Office-Home (Venkateswara et al., 2017), Terra Incognita (Beery et al., 2018) and DomainNet (Peng et al., 2019) which are the common DG benchmarks with multi-source domains.

- **PACS** (Li et al., 2017): 9991 images of seven classes in total, over four domains:Art_painting (A), Cartoon (C), Sketches (S), and Photo (P).
- **VLCS** (Torralba & Efros, 2011): five classes over four domains with a total of 10729 samples. The domains are defined by four image origins, i.e., images were taken from the PASCAL VOC 2007 (V), LabelMe (L), Caltech (C) and Sun (S) datasets.
- **Office-Home** (Venkateswara et al., 2017): 65 categories of 15500 daily objects from 4 domains: Art, Clipart, Product (vendor website with white-background) and Real-World (real-object collected from regular cameras).
- **Terra Incognita** (Beery et al., 2018) includes 24788 wild photographs of dimension (3, 224, 224) with 10 animals, over 4 camera-trap domains L100, L38, L43 and L46. This dataset contains photographs of wild animals taken by camera traps; camera trap locations are different across domains.
- **DomainNet** (Peng et al., 2019) contains 596006 images of dimension (3, 224, 224) and 345 classes, over 6 domains clipart, infograph, painting, quickdraw, real and sketch. This is the biggest dataset in terms of the number of samples and classes.

### 7.2 BASELINES

This appendix provides an literature review about 14 related domain generalization methods from DomainBed benchmark which are used to make comparisons with our model:

- **ERM** (Vapnik, 1999):minimizes the sum of errors across domains and examples. For our experiments, we employ the implementation from (Gulrajani & Lopez-Paz, 2021), a strong baseline that can achieve competitive accuracies on DG benchmarks.
- **IRM** (Arjovsky et al., 2019): learns invariant feature representation such that the optimal linear classifier on top of that representation matches across domains.
- **Mixup** (Wang et al., 2020): applies ERM on linear interpolations of examples from random pairs of domains and their labels.
- **CORAL** (Sun & Saenko, 2016): aligns training domain distributions at a specific level of representation by matching the mean and covariance, which are second-order statistics, of the features across these domains.
- **MMD** (Li et al., 2018b): employs the adversarial technique and the maximum mean discrepancy criteria to align latent distribution across domain.

- **MLDG** (Li et al., 2018a): proposed meta-learning strategy that splits meta train/test and performs gradient alignment to update each minibatch.

- **DANN** (Ganin et al., 2016): uses an adversarial network to align latent representation across domains.

- **CDANN** (Li et al., 2018b): is a variant of DANN that facilitates the alignment of multimodal distributions by matching the feature conditional distributions across domains for all class labels..

- **VREx** (Krueger et al., 2021): approximates IRM to reduce the variance of error averages across domains.

- **GroupDRO** (Sagawa et al., 2019): applies ERM while enhancing the significance of domains by assigning weights to mini-batches from the training distribution in proportion to their larger errors..

- **MTL** (Blanchard et al., 2021): estimating a kernel mean embedding for each domain, which is subsequently provided as a second argument to the classifier. These embeddings are estimated using individual test examples during the testing phase.

- **ARM** (Zhang et al., 2020): extending MTL by utilizing of a distinct Convolutional Neural Network (CNN) for the computation of domain embedding. This domain embedding is subsequently appended to the input images as supplementary channels.

- **SagNets** (Nam et al., 2021): mitigates the domain gap by incentivizing latent representations to disregard image style and prioritize content emphasis.

- **RSC** (Huang et al., 2020): iteratively dropping out the most activated features to challenge the network.

### 7.3 EXPERIMENTAL RESULT DETAILS

In this section, we show detailed results of Table 1 in the main text. Standard errors are reported from three trials.

### 7.3.1 VLCS

Table 4: Classification Accuracy on VLCS using ResNet50

| Algorithm | C | L | S | V | Avg |
|---|---|---|---|---|---|
| ERM | $97.7 \pm 0.4$ | $64.3 \pm 0.9$ | $73.4 \pm 0.5$ | $74.6 \pm 1.3$ | 77.5 |
| IRM | $98.6 \pm 0.1$ | $64.9 \pm 0.9$ | $73.4 \pm 0.6$ | $77.3 \pm 0.9$ | 78.5 |
| GroupDRO | $97.3 \pm 0.3$ | $63.4 \pm 0.9$ | $69.5 \pm 0.8$ | $76.7 \pm 0.7$ | 76.7 |
| Mixup | $98.3 \pm 0.6$ | $64.8 \pm 1.0$ | $72.1 \pm 0.5$ | $74.3 \pm 0.8$ | 77.4 |
| MLDG | $97.4 \pm 0.2$ | $65.2 \pm 0.7$ | $71.0 \pm 1.4$ | $75.3 \pm 1.0$ | 77.2 |
| CORAL | $98.3 \pm 0.1$ | $\mathbf{66.1} \pm 1.2$ | $73.4 \pm 0.3$ | $77.5 \pm 1.2$ | **78.8** |
| MMD | $97.7 \pm 0.1$ | $64.0 \pm 1.1$ | $72.8 \pm 0.2$ | $75.3 \pm 3.3$ | 77.5 |
| DANN | $99.0 \pm 0.3$ | $65.1 \pm 1.4$ | $73.1 \pm 0.3$ | $77.2 \pm 0.6$ | 78.6 |
| CDANN | $97.1 \pm 0.3$ | $65.1 \pm 1.2$ | $70.7 \pm 0.8$ | $77.1 \pm 1.5$ | 77.5 |
| MTL | $97.8 \pm 0.4$ | $64.3 \pm 0.3$ | $71.5 \pm 0.7$ | $75.3 \pm 1.7$ | 77.2 |
| SagNet | $97.9 \pm 0.4$ | $64.5 \pm 0.5$ | $71.4 \pm 1.3$ | $\underline{77.5} \pm 0.5$ | 77.8 |
| ARM | $98.7 \pm 0.2$ | $63.6 \pm 0.7$ | $71.3 \pm 1.2$ | $76.7 \pm 0.6$ | 77.6 |
| VREx | $98.4 \pm 0.3$ | $64.4 \pm 1.4$ | $\underline{74.1} \pm 0.4$ | $76.2 \pm 1.3$ | 78.3 |
| RSC | $97.9 \pm 0.1$ | $62.5 \pm 0.7$ | $72.3 \pm 1.2$ | $75.6 \pm 0.8$ | 77.1 |
| KEAI | $\underline{98,6} \pm 0.1$ | $\underline{65,6} \pm 1,0$ | $\mathbf{74,1} \pm 0.1$ | $\mathbf{79,3} \pm 0,3$ | **79,4** |

### 7.3.2 PACS

Table 5: Classification Accuracy on PACS using ResNet50

| Algorithm | A | C | P | S | Avg |
|---|---|---|---|---|---|
| ERM | $84.7 \pm 0.4$ | $\underline{80.8} \pm 0.6$ | $97.2 \pm 0.3$ | $79.3 \pm 1.0$ | 85.5 |
| IRM | $84.8 \pm 1.3$ | $76.4 \pm 1.1$ | $96.7 \pm 0.6$ | $76.1 \pm 1.0$ | 83.5 |
| GroupDRO | $83.5 \pm 0.9$ | $79.1 \pm 0.6$ | $96.7 \pm 0.3$ | $78.3 \pm 2.0$ | 84.4 |
| Mixup | $86.1 \pm 0.5$ | $78.9 \pm 0.8$ | $\mathbf{97.6} \pm 0.1$ | $75.8 \pm 1.8$ | 84.6 |
| MLDG | $85.5 \pm 1.4$ | $80.1 \pm 1.7$ | $97.4 \pm 0.3$ | $76.6 \pm 1.1$ | 84.9 |
| CORAL | $\mathbf{88.3} \pm 0.2$ | $80.0 \pm 0.5$ | $97.5 \pm 0.3$ | $78.8 \pm 1.3$ | 86.2 |
| MMD | $86.1 \pm 1.4$ | $79.4 \pm 0.9$ | $96.6 \pm 0.2$ | $76.5 \pm 0.5$ | 84.6 |
| DANN | $86.4 \pm 0.8$ | $77.4 \pm 0.8$ | $97.3 \pm 0.4$ | $73.5 \pm 2.3$ | 83.6 |
| CDANN | $84.6 \pm 1.8$ | $75.5 \pm 0.9$ | $96.8 \pm 0.3$ | $73.5 \pm 0.6$ | 82.6 |
| MTL | $87.5 \pm 0.8$ | $77.1 \pm 0.5$ | $96.4 \pm 0.8$ | $77.3 \pm 1.8$ | 84.6 |
| SagNet | $87.4 \pm 1.0$ | $80.7 \pm 0.6$ | $97.1 \pm 0.1$ | $\underline{80.0} \pm 0.4$ | $\underline{86.3}$ |
| ARM | $86.8 \pm 0.6$ | $76.8 \pm 0.5$ | $97.4 \pm 0.3$ | $79.3 \pm 1.2$ | 85.1 |
| VREx | $86.0 \pm 1.6$ | $79.1 \pm 0.6$ | $96.9 \pm 0.5$ | $77.7 \pm 1.7$ | 84.9 |
| RSC | $85.4 \pm 0.8$ | $79.7 \pm 1.8$ | $\underline{97.6} \pm 0.3$ | $78.2 \pm 1.2$ | 85.2 |
| KEAI | $\underline{87.4} \pm 0.2$ | $\mathbf{82.0} \pm 0.9$ | $97.5 \pm 0.1$ | $\mathbf{80.2} \pm 0.4$ | $\mathbf{86.8}$ |

### 7.3.3 OFFICEHOME

Table 6: Classification Accuracy on OfficeHome using ResNet50

| Algorithm | A | C | P | R | Avg |
|---|---|---|---|---|---|
| ERM | $61.3 \pm 0.7$ | $52.4 \pm 0.3$ | $75.8 \pm 0.1$ | $76.6 \pm 0.3$ | 66.5 |
| IRM | $58.9 \pm 2.3$ | $52.2 \pm 1.6$ | $72.1 \pm 2.9$ | $74.0 \pm 2.5$ | 64.3 |
| GroupDRO | $60.4 \pm 0.7$ | $52.7 \pm 1.0$ | $75.0 \pm 0.7$ | $76.0 \pm 0.7$ | 66.0 |
| Mixup | $62.4 \pm 0.8$ | $\underline{54.8} \pm 0.6$ | $\mathbf{76.9} \pm 0.3$ | $78.3 \pm 0.2$ | 68.1 |
| MLDG | $61.5 \pm 0.9$ | $53.2 \pm 0.6$ | $75.0 \pm 1.2$ | $77.5 \pm 0.4$ | 66.8 |
| CORAL | $\mathbf{65.3} \pm 0.4$ | $54.4 \pm 0.5$ | $\underline{76.5} \pm 0.1$ | $\mathbf{78.4} \pm 0.5$ | $\mathbf{68.7}$ |
| MMD | $60.4 \pm 0.2$ | $53.3 \pm 0.3$ | $74.3 \pm 0.1$ | $77.4 \pm 0.6$ | 66.3 |
| DANN | $59.9 \pm 1.3$ | $53.0 \pm 0.3$ | $73.6 \pm 0.7$ | $76.9 \pm 0.5$ | 65.9 |
| CDANN | $61.5 \pm 1.4$ | $50.4 \pm 2.4$ | $74.4 \pm 0.9$ | $76.6 \pm 0.8$ | 65.8 |
| MTL | $61.5 \pm 0.7$ | $52.4 \pm 0.6$ | $74.9 \pm 0.4$ | $76.8 \pm 0.4$ | 66.4 |
| SagNet | $63.4 \pm 0.2$ | $\mathbf{54.8} \pm 0.4$ | $75.8 \pm 0.4$ | $78.3 \pm 0.3$ | 68.1 |
| ARM | $58.9 \pm 0.8$ | $51.0 \pm 0.5$ | $74.1 \pm 0.1$ | $75.2 \pm 0.3$ | 64.8 |
| VREx | $60.7 \pm 0.9$ | $53.0 \pm 0.9$ | $75.3 \pm 0.1$ | $76.6 \pm 0.5$ | 66.4 |
| RSC | $60.7 \pm 1.4$ | $51.4 \pm 0.3$ | $74.8 \pm 1.1$ | $75.1 \pm 1.3$ | 65.5 |
| KEAI | $\underline{64,7} \pm 0.2$ | $54,1 \pm 0.6$ | $76,3 \pm 1.0$ | $\underline{78,4} \pm 0.8$ | $\underline{68,4}$ |

### 7.3.4 TERRAINCOGNITA

Table 7: Classification Accuracy on TerraIncognita using ResNet50

| Algorithm | L100 | L38 | L43 | L46 | Avg |
|-----------|------|-----|-----|-----|-----|
| ERM | 49.8 ± 4.4 | 42.1 ± 1.4 | 56.9 ± 1.8 | 35.7 ± 3.9 | 46.1 |
| IRM | 54.6 ± 1.3 | 39.8 ± 1.9 | 56.2 ± 1.8 | 39.6 ± 0.8 | 47.6 |
| GroupDRO | 41.2 ± 0.7 | 38.6 ± 2.1 | 56.7 ± 0.9 | 36.4 ± 2.1 | 43.2 |
| Mixup | **59.6** ± 2.0 | 42.2 ± 1.4 | 55.9 ± 0.8 | 33.9 ± 1.4 | 47.9 |
| MLDG | 54.2 ± 3.0 | **44.3** ± 1.1 | 55.6 ± 0.3 | 36.9 ± 2.2 | 47.7 |
| CORAL | 51.6 ± 2.4 | 42.2 ± 1.0 | 57.0 ± 1.0 | 39.8 ± 2.9 | 47.6 |
| MMD | 41.9 ± 3.0 | 34.8 ± 1.0 | 57.0 ± 1.9 | 35.2 ± 1.8 | 42.2 |
| DANN | 51.1 ± 3.5 | 40.6 ± 0.6 | 57.4 ± 0.5 | 37.7 ± 1.8 | 46.7 |
| CDANN | 47.0 ± 1.9 | 41.3 ± 4.8 | 54.9 ± 1.7 | 39.8 ± 2.3 | 45.8 |
| MTL | 49.3 ± 1.2 | 39.6 ± 6.3 | 55.6 ± 1.1 | 37.8 ± 0.8 | 45.6 |
| SagNet | 53.0 ± 2.9 | 43.0 ± 2.5 | **57.9** ± 0.6 | 40.4 ± 1.3 | **48.6** |
| ARM | 49.3 ± 0.7 | 38.3 ± 2.4 | 55.8 ± 0.8 | 38.7 ± 1.3 | 45.5 |
| VREx | 48.2 ± 4.3 | 41.7 ± 1.3 | 56.8 ± 0.8 | 38.7 ± 3.1 | 46.4 |
| RSC | 50.2 ± 2.2 | 39.2 ± 1.4 | 56.3 ± 1.4 | 40.8 ± 0.6 | 46.6 |
| KEAI | 52,6 ± 7.0 | 40,6 ± 1.2 | 57,3 ± 0.1 | **43,8** ± 0,7 | **48,6** |

### 7.3.5 DOMAINNET

Table 8: Classification Accuracy on DomainNet using ResNet50

| Algorithm | clip | info | paint | quick | real | sketch | Avg |
|-----------|------|------|-------|-------|------|--------|-----|
| ERM | 58.1 ± 0.3 | 18.8 ± 0.3 | 46.7 ± 0.3 | 12.2 ± 0.4 | 59.6 ± 0.1 | 49.8 ± 0.4 | 40.9 |
| IRM | 48.5 ± 2.8 | 15.0 ± 1.5 | 38.3 ± 4.3 | 10.9 ± 0.5 | 48.2 ± 5.2 | 42.3 ± 3.1 | 33.9 |
| GroupDRO | 47.2 ± 0.5 | 17.5 ± 0.4 | 33.8 ± 0.5 | 9.3 ± 0.3 | 51.6 ± 0.4 | 40.1 ± 0.6 | 33.3 |
| Mixup | 55.7 ± 0.3 | 18.5 ± 0.5 | 44.3 ± 0.5 | 12.5 ± 0.4 | 55.8 ± 0.3 | 48.2 ± 0.5 | 39.2 |
| MLDG | 59.1 ± 0.2 | 19.1 ± 0.3 | 45.8 ± 0.7 | **13.4** ± 0.3 | 59.6 ± 0.2 | 50.2 ± 0.4 | 41.2 |
| CORAL | 59.2 ± 0.1 | 19.7 ± 0.2 | 46.6 ± 0.3 | 13.4 ± 0.4 | **59.8** ± 0.2 | 50.1 ± 0.6 | 41.5 |
| MMD | 32.1 ± 13.3 | 11.0 ± 4.6 | 26.8 ± 11.3 | 8.7 ± 2.1 | 32.7 ± 13.8 | 28.9 ± 11.9 | 23.4 |
| DANN | 53.1 ± 0.2 | 18.3 ± 0.1 | 44.2 ± 0.7 | 11.8 ± 0.1 | 55.5 ± 0.4 | 46.8 ± 0.6 | 38.3 |
| CDANN | 54.6 ± 0.4 | 17.3 ± 0.1 | 43.7 ± 0.9 | 12.1 ± 0.7 | 56.2 ± 0.4 | 45.9 ± 0.5 | 38.3 |
| MTL | 57.9 ± 0.5 | 18.5 ± 0.4 | 46.0 ± 0.1 | 12.5 ± 0.1 | 59.5 ± 0.3 | 49.2 ± 0.1 | 40.6 |
| SagNet | 57.7 ± 0.3 | 19.0 ± 0.2 | 45.3 ± 0.3 | 12.7 ± 0.5 | 58.1 ± 0.5 | 48.8 ± 0.2 | 40.3 |
| ARM | 49.7 ± 0.3 | 16.3 ± 0.5 | 40.9 ± 1.1 | 9.4 ± 0.1 | 53.4 ± 0.4 | 43.5 ± 0.4 | 35.5 |
| VREx | 47.3 ± 3.5 | 16.0 ± 1.5 | 35.8 ± 4.6 | 10.9 ± 0.3 | 49.6 ± 4.9 | 42.0 ± 3.0 | 33.6 |
| RSC | 55.0 ± 1.2 | 18.3 ± 0.5 | 44.4 ± 0.6 | 12.2 ± 0.2 | 55.7 ± 0.7 | 47.8 ± 0.9 | 38.9 |
| KEAI | **59,9** ± 0,3 | **20,9** ± 0.6 | **49,1** ± 0.2 | 12,0 ± 0.6 | 59,8 ± 1.2 | **50,8** ± 0.6 | **42,1** |

## 8 EXPERIMENTAL SETTINGS

### 8.1 IMPLEMENTATION DETAILS

we adopted a simple and effective strategy for dividing the latent vector, aimed at facilitating a fair comparison. This method entails segmenting the latent vector into several sub-features, while consciously avoiding the introduction of extra modules or transformations that could modify its fundamental structure. For ease of implementation, we opted to split the latent vector into attributes of equal dimensions.

We detail our algorithm's implementation:

- *Feature extractor network*: we adopt the ResNet50 (He et al., 2016) architecture (removing the final classification layer) and the batch normalization statistics are frozen during training.

- *Kernel-based Feature-Identifier* $\Gamma$: each *Kernel-based Attribute Identifier* $\Gamma_k$ is modelled as a max-margin hyperplane $(W_k, b_k)$, having equation $W_k^\top \phi_\sigma(a) + b_k = 0$ where $(w_k, b_k)$ is implemented by i.e., a Full-Connected (FC) layer $n_a \times 1$, attribute $a \in \mathbb{R}^{n_a}$ is the input and $\phi_\sigma : \mathcal{U} \to \mathbb{R}^{2 \times D}$ is the feature map from attribute space to kernel space defined as Eq.4 in the main paper.

## 8.2 HYPERPARAMETERS

The hyperparameters are chosen using the same strategy as in (Cha et al., 2021). Specifically, we employ the Adam optimizer as described in (Gulrajani & Lopez-Paz, 2021), with a learning rate in the search range of $[1e^{-5}, 5e^{-5}]$ and no dropout or weight decay. The batch size is set to 32 and the attribute-dim is in the search choice of $n_a \in \{32, 64, 128\}$ for all datasets in our main experiments. The number of total iterations is 15,000 for DomainNet and 5,000 for other datasets, which are considered sufficient for convergence. As for the kernel hyperparameters, we adopt the initial values of $\sigma = 1.0$, and $D = 128$ based on insights from (Nguyen et al., 2017). These values remain consistent across all experiments. Finally, we use system of NVIDIA QUADRO RTX 6000 to conduct our experiments.

## 8.3 COMPUTATIONAL COST

In our KEAI framework, we divide the latent space into $K$ groups and implement a classifier and a kernel model for each attribute. This results in $K$ attribute-based classifiers and $K$ kernel models $\{\Gamma_k\}_1^K$. It's important to highlight that each attribute-based classifier processes an input attribute with a dimension of $n_a = \frac{n_z}{K}$. As such, the cumulative parameters for the $K$ linear attribute-based classifiers are equivalent to those of a single linear classifier with an input dimension of $n_z$. We utilize batch-matrix computation (as detailed in Listing.1) to simultaneously forward all $K$ linear attribute-based classifiers. Consequently, the total parameter count and computational load are comparable to a standard linear classifier operating on the original latent space.

A similar approach is adopted for the $K$ kernel models $\{\Gamma_k\}_1^K$, where the hyperplanes are also modeled as linear layers. In summary, the additional computation involves only the transformation from attribute space to kernel space as per Eq.4 and a linear classifier for the hyperplanes.

```python
class EnsembleLinear(nn.Linear):
    def __init__(self, ensemble_size, attribute_dim, output_dim):
        nn.Module.__init__(self)
        self.attribute_dim = attribute_dim
        self.output_dim = output_dim
        self.weight = nn.Parameter(torch.Tensor(ensemble_size,
    attribute_dim, output_dim))
        self.bias = nn.Parameter(torch.Tensor(ensemble_size, 1,
    output_dim))

    def forward(self, x):
        # x: Ensemble Batch Attribute_in
        return torch.baddbmm(self.bias, x, self.weight)
```
Listing 1: Ensemble Linear Layer

