# OpenReview forum: "KEFI: Kernel-based Feature Identification for Generalizable Classification"
_ICLR.cc/2024/Conference — Submitted to ICLR 2024_

### Official Review · Reviewer_TEu1 · 2023-10-26

**Soundness:** 2 fair
**Presentation:** 2 fair
**Contribution:** 2 fair
**Rating:** 3
**Confidence:** 3

**Summary:**

This paper studies the domain-free domain generalization (DG) problem, where the domain labels for multiple source domains are unavailable in the learning procedure. To extract the representations with explicit semantic structure and interpretable components, i.e., attributes, this work proposes a kernel-based attributes identifier, where the attributes are learned with parameterized feature extractors and then the effective attributes are selected by the proposed identifier. By feature selection, the learned representations have less redundant information compared with the considering the combination of all attributes. Experiments on standard DG datasets are conducted, where the proposed method outperforms other comparison methods.

**Strengths:**

+ A kernel-based feature extraction and selection method is proposed to learn the attributes in multiple domains; the basic motivation is clear.
+ Experiment analysis is conducted from different aspects, e.g., comparison and ablation.

**Weaknesses:**

- The related works in DG are not properly discussed. Indeed, there are methods that share the same motivation and goal with the proposed method, i.e., learning attributes with the complex multi-domain data. Thus, it is hard to evaluate the real merits of this paper.
- The technical parts are presented directly, while the innovations and justifications are insufficient.
- The comparison methods in experiments should contain more latest SOTA methods.

**Questions:**

1. The discussion on related works is indeed insufficient. Basically, this method focuses on learning interpretable representations with explicitly decomposed components. However, there are works that also consider the decomposition or encoding of attributes from different perspectives, e.g., information bottleneck [r1] and rate reduction [r2].

2. In the methodology part, only the kernel-based feature selection procedure is presented, while the justification and in-depth analysis are also necessary. Specifically, compared with other methods that also aim at learning attributes, what are the limitations in the existing methodology? Correspondingly, what are the advantages and weaknesses of the proposed method in learning components?

3. In experiment validation, the proposed method achieves higher accuracies compared with other methods, while the comparison methods are generally proposed before 2021. To ensure a fair and appropriate comparison, the latest SOTA methods in DG are highly expected to be considered.

4. Though it is natural to decompose the input into different attributes, the number of attributions $K$ seems to be a hyper-parameter and the learned $K$ attributes lack interpretability. In related works, e.g., [r1], some justifications and intuitions are provided for learned attributes.

5. Since $K$ is a hyper-parameter, it is indeed necessary to provide theoretical analysis or reasonable explanation. In its current form, only experiment results with different parameter selection is provided, while the analysis is insufficient.


[r1] Li, Bo, et al. "Invariant information bottleneck for domain generalization." Proceedings of the AAAI Conference on Artificial Intelligence. Vol. 36. No. 7. 2022.

[r2] Chan, Kwan Ho Ryan, et al. "ReduNet: A white-box deep network from the principle of maximizing rate reduction." The Journal of Machine Learning Research 23.1 (2022): 4907-5009.

---

> ### Author Response · Authors · 2023-11-21
>
> We sincerely appreciate your thorough and insightful feedback. Below, we outline our efforts to address the valuable points you have raised.
>
> &nbsp;
>
> **The discussion on related works is indeed insufficient. Basically, this method focuses on learning interpretable representations with explicitly decomposed components. However, there are works that also consider the decomposition or encoding of attributes from different perspectives, e.g., information bottleneck [r1] and rate reduction [r2].**
>
> *Reply:*
>
> In response to [r1, r2], both methods are centered on learning invariant representations, which stands in contrast to our approach that focuses on capturing a broad spectrum of features from source domains. Specifically, [r1] promotes the learning of invariant representations by integrating the invariant risk constraint of IRM and a sparsity constraint into the objective function, through the lens of mutual information, a concept they term the Invariant Information Bottleneck (IIB) principle. On the other hand, [r2] aims to find an optimal (invariant) linear discriminative representation of data by optimizing the rate reduction objective.
>
> &nbsp;
>
> **Discussion on related works:**
>
> In term of feature decomposition, It is difficult to find research with analogous characteristics to our method:
> - (i) capture a wide range of attributes from source domains
> -  (ii) provide a machanism for Attribute-Identifiability, essential for discerning if an attribute in the target domain has been previously encountered in the source domain.
>
> While existing methods e.g, [1,2,3] in the literature aim to learn a variety of features, they generally utilize all learned attributes uniformly across various target domains, without distinction.
>
> &nbsp;
>
> In term of kernel-based method for DG, we have expanded the related work section to include additional kernel-based methods, highlighted in blue, and provided a discussion contrasting our approach with these previous methods as follows:
>
> *Diverging from these existing methods, our proposed approach also falls under kernel-based methods but takes a distinct route. Unlike traditional techniques that utilize the kernel as a tool for measuring divergence or similarity and rely on the kernel trick to avoid explicit computation of the feature map, our method takes a direct approach. We compute the feature map through a random finite-dimensional feature map. This enables us to leverage a clustering-based perspective of kernel methods, which is instrumental in characterizing and refining representations for effective domain generalization.*
>
> &nbsp;
>
> [1] Huang, Zeyi, et al. "Self-challenging improves cross-domain generalization." ECCV 2020
>
> [2] Bui, Manh-Ha, et al. "Exploiting domain-specific features to enhance domain generalization." NIPS 2021
>
> [3] Chattopadhyay, Prithvijit, , et al. "Learning to balance specificity and invariance for in and out of domain generalization." ECCV 2020
>
>
> &nbsp;
>
> **In the methodology part, only the kernel-based feature selection procedure is presented, while the justification and in-depth analysis are also necessary. Specifically, compared with other methods that also aim at learning attributes, what are the limitations in the existing methodology? Correspondingly, what are the advantages and weaknesses of the proposed method in learning components?**
>
> *Reply:*
>
> Our approach includes a mechanism for Attribute-Identifiability, which is crucial for determining whether an attribute in the target domain was previously present in the source domain. However, it's important to note that this is not solely a kernel-based feature selection process.
>
> To elaborate further, we would like to offer additional insights into the fundamental aspects of our methodology:
>
> - Attribute-based representations are often expected to be easily interpretable and meaningful to the users of the model. Typically, learning and evaluating such models necessitates specialized domain knowledge. In our research, we reduce the need for such in-depth expertise by aligning our work with a specific downstream task: classification, as outlined in Definition 1.
>
> - We have developed a thorough framework for attribute identification, framing it as a kernel-based optimization problem (Section 3.4). In particular, we utilize the clustering perspective of kernel methods for attribute identification. To the best of our knowledge, no prior studies have applied the cluster-induced viewpoint of kernel methods at the attribute level for addressing domain generalization. Our work is pioneering in exploring this novel approach.
>
> - Beyond just identifying bases of attributes, our sophisticated formulation also promotes the distinctness of attribute clusters (**Remark 1**). This approach effectively enhances the discriminative power of the attributes we learn, contributing to the overall robustness and efficacy of our model.

---

> > ### Author Response · Authors · 2023-11-21
> >
> > In this work, our primary interest lies in exploring the properties of \textit{kernel-based attributes} and their advantages, rather than pursuing state-of-the-art performance.
> >
> > Recent approaches, such as [1, 2, 3, 4], frequently adopt an ensemble strategy, particularly one based on SWAD [1]. This approach complicates the evaluation of the effectiveness of their latent representations. As a result, our research primarily focuses on methods that rely on a singular model for evaluation. Nonetheless, to provide a comprehensive view, we have broadened our related works section to include these methodologies. Additionally, we have conducted extra experiments where our KEAI model is trained using an ensemble strategy similar to SWAD, as shown in the following table.
> >
> > These experiments reveal that KEAI not only performs competitively when compared to state-of-the-art (SOTA) methods but also shows notable effectiveness in domains with limited information, such as the Cartoon and Sketch domains.
> >
> > &nbsp;
> >
> > ***Table 1: Classification Accuracy on PACS with ResNet18***
> >
> > | Algorithm  | Photo | Art-painting | Cartoon | Sketch | Average |
> > | :---- | :----: | :----: | :----: | :----: |----: |
> > |SWAD[1] | 89.3 +/-0.2 | 83.4 +/-0.6 | 97.3 +/-0.3 | 82.5 +/-0.5 | 88.1|
> > |DNA[2] | 89.8 +/-0.2 | 83.4 +/-0.4 | 97.7 +/-0.1 | 82.6 +/-0.2 | 88.4|
> > |PCL [3]| **90.2 +/-0.0** | 83.9 +/-0.0 | **98.1 +/-0.0** | 82.6 +/-0.0 | **88.7**|
> > |VNE[4] |90.1 +/-0.0 | 83.8 +/-0.0  | 97.5 +/-0.0 |  81.8 +/-0.0 |  88.3|
> > |KEAI | 89.2+/-0.3| **84.2+/-0.7**| 97.3+/-0.3 | **83.1+/-0.7**| 88.5|
> >
> > &nbsp;
> >
> > [1] Junbum, Cha et al. (2021). “SWAD: domain generalization by seeking flat minima.” In: NeurIPS.
> >
> >
> > [2] Xu, Chu et al. (2022). “DNA: domain generalization with diversified neural averaging.” In: ICML.
> >
> > [3] Pcl: Proxy-based contrastive learning for domain generalization, CVPR2022.
> >
> > [4] Kim, Jaeill, et al. "VNE: An Effective Method for Improving Deep Representation by Manipulating Eigenvalue Distribution." Proceedings of the IEEE/CVF Conference on Computer Vision and Pattern Recognition. 2023.

---

### Official Review · Reviewer_ysfS · 2023-10-31

**Soundness:** 3 good
**Presentation:** 3 good
**Contribution:** 3 good
**Rating:** 5
**Confidence:** 3

**Summary:**

The paper proposes an attribute-based feature extractor and leveraging kernel learning theory to delineate the decision region of attributes collected from the source domain.

**Strengths:**

1. The KEAI framework efficiently captures meaningful components from source domains, addressing challenges in domain generalization.

**Weaknesses:**

1. How to find K (K group), by tuning? If yes, the ablation study on K is missing
2. Why are the dimensions of attributes equal? An explanation or proof is missing
3. The computation cost of the whole algorithm is missing
4. Experimental results are not convincing. E.g., Table 1,  the experimental results seem sometimes worse than others. Ablation study didn't compared with other methods.
5. Some typo in formulations, e.g., missing +C*$\xi_k$ in soft version

**Questions:**

1. Confused about the basis of attributes? How could you find the basis of attributes by formulation (3) without any attribute labels?

---

> ### Author Response · Authors · 2023-11-20
> **Response to Reviewer ysfS (1/2)**
>
> We sincerely appreciate your thorough and insightful feedback. Below, we outline our efforts to address the valuable points you have raised.
>
> &nbsp;
>
> **Confused about the basis of attributes?**
>
> *Reply:*
>
> In our study, we initially segment the latent space into $K$ groups, where all attributes (sub-latent vectors) within a single group form a basis. Typically, attribute-based representations are generally presumed to be interpretable and meaningful for the end-users of the model. Typically. Learning and evaluating such models require domain-specific knowledge. In our work, we mitigate this requirement for domain expertise in attribute-representation by focusing on a downstream task, specifically classification, as follows:
>
> *"A basis of attributes in relation to a given dataset is defined in terms that the attributes within a basis should be capable of predicting the corresponding class-labels for all samples in the dataset."*
>
> Formal definition:
>
> **(Basis of attributes)**: For a dataset $\mathbb{D}=\\{(x^i,y^i)\\}_{i=1}^N$ and feature extractor $g$, a set $\mathcal{A}\_{k}=\\{ a^1_k,a^2_k,...,a^{N}_k \\\}$ where each $a^i_k=g(x^i)_k$ is considered as *basis of attributes* with respect to $(\mathbb{D},g)$ if there exists a function $h_k$ such that $h_k(a^i_k)=y_i, \forall i=1...N$.
>
> This constraints can be obtained by optimize objective function in Equation.(3): training attribute-based classifiers in source domains.
>
> &nbsp;
>
> **How could you find the basis of attributes by formulation (3) without any attribute labels?**
>
> *Reply:*
>
> We formulate the attribute detection in the target domain as an outlier detection problem.
>
> We start with cluster induced representation of kernel method.
> The hyperplane on kernel space when projecting back into the original attribute space, resulting in a series of contours that envelop the target attribute. These contours serve as cluster boundaries, with points within each distinct contour being assigned to the same cluster.  Additionally, it is found that these contours effectively outline the support of the underlying probability distribution, essentially highlighting high-density regions in the distribution of target attribute. This characteristic is particularly valuable for outlier detection.
>
> During the training process, as we divide the latent space into $K$ groups, all attributes within the same group constitute a basis. To learn the cluster boundaries (hyperplane) $\Gamma_k$ for a basis, we treat attributes from other bases as negatives.
>
> Referring Section 3.4:
>
> *Specifically, the max-margin hyperplane $\Gamma_{k}$ is built in such way that:
> (i) it separates push-forward kernel-features of positive attributes  $C\_{+k}=\\{\phi\_{\sigma}(a) \\mid a\in\mathcal{A}\_{k}\\}$ and the negative attributes $C\_{-k}=\\{\phi\_{\sigma}(a)\mid a\in\mathcal{A}\_{i\neq k}\\}$ and (ii) the margin w.r.t the hyperplane $\Gamma_{k}$ defined as the closest distance from negative in $C_{-k}$ to this hyperplane is maximized.*
>
> It is important to note that our hyperplane in the kernel feature space is integrated with the encoder $g$. Hence, optimizing hyperlanes on kernel space also updates representation of attributes. This integration allows for a more compact refinement of attribute clusters from the basis $A_k$, while also efficiently distinguishing them from attributes derived from other bases.
>
> Upon completion of training, we obtain a set of hyperplanes $\\{\Gamma_k\\}_1^K$, which function as tools for outlier detection. These hyperplanes help in determining whether an attribute in the target domain was learned in the source domain.
>
> &nbsp;
>
> **How to find K (K group), by tuning? If yes, the ablation study on K is missing**
>
> *Reply:*
>
> In current work, K is Hyper-parameters. We have provided ablation study with different value of $K$ in Section 4.3.2.
>
> &nbsp;
>
> **Why are the dimensions of attributes equal? An explanation or proof is missing**
>
> *Reply:*
>
> In our study, we adopted a simple and effective strategy for dividing the latent vector, aimed at facilitating a fair comparison. This method entails segmenting the latent vector into several sub-features, while consciously avoiding the introduction of extra modules or transformations that could modify its fundamental structure. For ease of implementation, we opted to split the latent vector into attributes of equal dimensions.
>
> This discussion has been added into Section 8.1: Implementation Details in the revised version (highlighted in blue).

---

> > ### Author Response · Authors · 2023-11-20
> >
> > **The computation cost of the whole algorithm is missing**
> >
> > *Reply:*
> >
> > We have added computational cost analysis and python implementation of our "esemble\_layer" in Appendix 8.3 as follows:
> >
> > In our KEAI framework, we divide the latent space into $K$ groups and implement a classifier and a kernel model for each attribute. This results in $K$ attribute-based classifiers and $K$ kernel models $\\{\Gamma_k\\}_1^K$. It's important to highlight that each attribute-based classifier processes an input attribute with a dimension of $n_a = \frac{n_z}{K}$. As such, the cumulative parameters for the $K$ linear attribute-based classifiers are equivalent to those of a single linear classifier with an input dimension of $n_z$. We utilize batch-matrix computation (as detailed in Appendix 8.3: Listing.1) to simultaneously forward all $K$ linear attribute-based classifiers. Consequently, the total parameter count and computational load are comparable to a standard linear classifier operating on the original latent space.
> >
> > A similar approach is adopted for the $K$ kernel models $\\{\Gamma_k\\}_1^K$, where the hyperplanes are also modeled as linear layers. In summary, the additional computation involves only the transformation from attribute space to kernel space as per Eq.(4) and a linear classifier for the hyperplanes.
> >
> > &nbsp;
> >
> > **Experimental results are not convincing. E.g., Table 1, the experimental results seem sometimes worse than others. Ablation study didn't compared with other methods.**
> >
> > In this work, our primary interest lies in exploring the properties of *kernel-based attributes* and their advantages, rather than pursuing state-of-the-art performance.
> >
> > Conducting an ablation study in comparison with other methods presents a significant challenge, as it is difficult to find research with analogous characteristics:
> > - (i) capture a wide range of attributes from source domains
> > - (ii) provide a machanism for Attribute-Identifiability, essential for discerning if an attribute in the target domain has been previously encountered in the source domain.
> >
> > While existing methods in the literature aim to learn a variety of features, they generally utilize all learned attributes uniformly across various target domains, without distinction.
> >
> > In term of kernel-based method for DG. Although our proposed approach also falls under kernel-based methods but takes a distinct route. Unlike traditional techniques that utilize the kernel as a tool for measuring divergence or similarity and rely on the kernel trick to avoid explicit computation of the feature map, our method takes a direct approach. We compute the feature map through a random finite-dimensional feature map. This enables us to leverage a clustering-based perspective of kernel methods, which is instrumental in characterizing and refining representations for effective domain generalization.
> >
> >
> > Additionally, recent approaches, such as [1, 2, 3, 4], frequently adopt an ensemble strategy, particularly one based on SWAD [1]. This approach complicates the evaluation of the effectiveness of their latent representations. As a result, our research primarily focuses on methods that rely on a singular model for evaluation. Nonetheless, to provide a comprehensive view, we have broadened our related works section to include these methodologies. Additionally, we have conducted extra experiments where our KEAI model is trained using an ensemble strategy similar to SWAD, as shown in the following table.
> >
> > These experiments reveal that KEAI not only performs competitively when compared to state-of-the-art (SOTA) methods but also shows notable effectiveness in domains with limited information, such as the Cartoon and Sketch domains.
> >
> > &nbsp;
> >
> > ***Table 1: Classification Accuracy on PACS with ResNet18***
> >
> > | Algorithm  | Photo | Art-painting | Cartoon | Sketch | Average |
> > | :---- | :----: | :----: | :----: | :----: |----: |
> > |SWAD[1] | 89.3 +/-0.2 | 83.4 +/-0.6 | 97.3 +/-0.3 | 82.5 +/-0.5 | 88.1|
> > |DNA[2] | 89.8 +/-0.2 | 83.4 +/-0.4 | 97.7 +/-0.1 | 82.6 +/-0.2 | 88.4|
> > |PCL [3]| **90.2 +/-0.0** | 83.9 +/-0.0 | **98.1 +/-0.0** | 82.6 +/-0.0 | **88.7**|
> > |VNE[4] |90.1 +/-0.0 | 83.8 +/-0.0  | 97.5 +/-0.0 |  81.8 +/-0.0 |  88.3|
> > |KEAI | 89.2+/-0.3| **84.2+/-0.7**| 97.3+/-0.3 | **83.1+/-0.7**| 88.5|
> >
> > &nbsp;
> >
> > [1] Junbum, Cha et al. (2021). “SWAD: domain generalization by seeking flat minima.” In: NeurIPS.
> >
> >
> > [2] Xu, Chu et al. (2022). “DNA: domain generalization with diversified neural averaging.” In: ICML.
> >
> > [3] Pcl: Proxy-based contrastive learning for domain generalization, CVPR2022.
> >
> > [4] Kim, Jaeill, et al. "VNE: An Effective Method for Improving Deep Representation by Manipulating Eigenvalue Distribution." Proceedings of the IEEE/CVF Conference on Computer Vision and Pattern Recognition. 2023.

---

### Official Review · Reviewer_N1P7 · 2023-11-01

**Soundness:** 3 good
**Presentation:** 2 fair
**Contribution:** 2 fair
**Rating:** 3
**Confidence:** 5

**Summary:**

This paper proposed an attribute-based feature extractor to capture semantically meaningful components  from the source domains referred using a Kernel-based Attribute Identifier. It leverages kernel learning theory to define the decision boundaries for these attributes collected from the source domains. They evaluate on multiple benchmarks to compare with several methods.

**Strengths:**

This paper proposed an attribute-based feature extractor to capture semantically meaningful components  from the source domains referred using a Kernel-based Attribute Identifier. Generally, the paper is easy to follow and such idea is interesting. They evaluate on multiple benchmarks to compare with several methods.

**Weaknesses:**

The attribute-based representation seems very tricky to obtain, especially for multiple source domains, it is not reasonable to put all sources together to learn the attribute-based representation. Different domains do have their own domain-specific attributes. The experiments do have provided such results to interpret what the attribute-based representation. Are they meaningful or matched with human-understandable attributes.

The performance improvement is not good enough, as shown in Table 1. The improvement is marginal.

**Questions:**

The interpretation of attribute-based representation.

The performance improvement.

---

> ### Author Response · Authors · 2023-11-20
> **Response to Reviewer N1P7 (1/2)**
>
> We sincerely appreciate your thorough and insightful feedback. Below, we outline our efforts to address the valuable points you have raised.
>
> &nbsp;
>
> **The interpretation of attribute-based representation.**
>
> In our study, we initially segment the latent space into $K$ groups, where all attributes (sub-latent vectors) within a single group form a basis.
>
> We concur with the reviewer's viewpoint that attribute-based representations are generally presumed to be interpretable and meaningful for the end-users of the model. Typically, learning and evaluating such models require domain-specific knowledge. In our work, we mitigate this requirement for domain expertise in attribute-representation by focusing on a downstream task, specifically classification, as follows:
>
> *"A basis of attributes in relation to a given dataset is defined in terms that the attributes within a basis should be capable of predicting the corresponding class-labels for all samples in the dataset."*
>
> Formal definition:
>
> **(Basis of attributes)**: For a dataset $\mathbb{D}=\\{(x^i,y^i)\\}_{i=1}^N$ and feature extractor $g$, a set $\mathcal{A}\_{k}=\\{ a^1_k,a^2_k,...,a^{N}_k \\\}$ where each $a^i_k=g(x^i)_k$ is considered as \textit{basis of attributes} with respect to $(\mathbb{D},g)$ if there exists a function $h_k$ such that $h_k(a^i_k)=y_i, \forall i=1...N$.
>
> This constraints can be obtained by optimize objective function in Equation.(3). However, we further employ cluster-induced viewpoint of kernel method to characterize attributes in bases. To learn the cluster boundaries (hyperplane) $\Gamma_k$ for a basis, we treat attributes from other bases as negatives.
>
> It is important to note that our hyperplane in the kernel feature space is integrated with the encoder $g$. Hence, optimizing hyperlanes on kernel space also updates representation of attributes. This integration allows for a more compact refinement of attribute clusters from the basis $A_k$, while also efficiently distinguishing them from attributes derived from other bases.
>
> Upon completion of training, we obtain a set of hyperplanes $\\{\Gamma_k\\}_1^K$, which function as tools for outlier detection. These hyperplanes help in determining whether an attribute in the target domain was learned in the source domain.
>
> In another interpretation, the basic of attributes finally are characterized by the kernel-models $\\{\Gamma_k\\}_1^K$, which we refer to as kernel-based attributes.
>
>
> &nbsp;
>
>
>
> ### **The performance improvement.**
>
> In this work, our primary interest lies in exploring the properties of \textit{kernel-based attributes} and their advantages, rather than pursuing state-of-the-art performance.
>
> Recent approaches, such as [1, 2, 3, 4], frequently adopt an ensemble strategy, particularly one based on SWAD [1]. This approach complicates the evaluation of the effectiveness of their latent representations. As a result, our research primarily focuses on methods that rely on a singular model for evaluation. Nonetheless, to provide a comprehensive view, we have broadened our related works section to include these methodologies. Additionally, we have conducted extra experiments where our KEAI model is trained using an ensemble strategy similar to SWAD, as shown in the following table.
>
> These experiments reveal that KEAI not only performs competitively when compared to state-of-the-art (SOTA) methods but also shows notable effectiveness in domains with limited information, such as the Cartoon and Sketch domains.
>
> &nbsp;
>
> ***Table 1: Classification Accuracy on PACS with ResNet18***
>
> | Algorithm  | Photo | Art-painting | Cartoon | Sketch | Average |
> | :---- | :----: | :----: | :----: | :----: |----: |
> |SWAD[1] | 89.3 +/-0.2 | 83.4 +/-0.6 | 97.3 +/-0.3 | 82.5 +/-0.5 | 88.1|
> |DNA[2] | 89.8 +/-0.2 | 83.4 +/-0.4 | 97.7 +/-0.1 | 82.6 +/-0.2 | 88.4|
> |PCL [3]| **90.2 +/-0.0** | 83.9 +/-0.0 | **98.1 +/-0.0** | 82.6 +/-0.0 | **88.7**|
> |VNE[4] |90.1 +/-0.0 | 83.8 +/-0.0  | 97.5 +/-0.0 |  81.8 +/-0.0 |  88.3|
> |KEAI | 89.2+/-0.3| **84.2+/-0.7**| 97.3+/-0.3 | **83.1+/-0.7**| 88.5|
>
> &nbsp;
>
> [1] Junbum, Cha et al. (2021). “SWAD: domain generalization by seeking flat minima.” In: NeurIPS.
>
>
> [2] Xu, Chu et al. (2022). “DNA: domain generalization with diversified neural averaging.” In: ICML.
>
> [3] Pcl: Proxy-based contrastive learning for domain generalization, CVPR2022.
>
> [4] Kim, Jaeill, et al. "VNE: An Effective Method for Improving Deep Representation by Manipulating Eigenvalue Distribution." Proceedings of the IEEE/CVF Conference on Computer Vision and Pattern Recognition. 2023.

---

> > ### Author Response · Authors · 2023-11-20
> > **Response to Reviewer N1P7 (2/2)**
> >
> > **Weaknesses:** *The attribute-based representation seems very tricky to obtain, especially for multiple source domains, it is not reasonable to put all sources together to learn the attribute-based representation. Different domains do have their own domain-specific attributes. The experiments do have provided such results to interpret what the attribute-based representation. Are they meaningful or matched with human-understandable attributes.*
> >
> > *Reply:*
> >
> > We believe that the assumption of invariant representation is essential for the efficacy of Domain Generalization (DG), though acquiring a truly invariant representation remains a significant challenge.
> >
> > Typically, latent representations can be categorized into three types of attributes, as disussed in [5]:
> >
> > - True invariant attributes: These are present across all source domains and the test domain.
> > - Pseudo-invariant attributes: Present in all source domains, but absent in the test domain.
> > - Domain-specific attributes: Found in one or several source domains, but not in the test domain.
> >
> > Guaranteeing that the learned attributes are true invariant attributes is nearly impossible. However, we can aim to learn pseudo-invariant attributes from the source domains. Therefore, we believe it is reasonable to capture a wide range of pseudo-invariant attributes by jointly optimizing Equation.(3): the pseudo-invariant constraint and Equation.(6): the diversity and identity constraint.
> >
> > &nbsp;
> >
> > [5] Li, Bo, et al. "Invariant information bottleneck for domain generalization." Proceedings of the AAAI Conference on Artificial Intelligence. Vol. 36. No. 7. 2022.

---

### Official Review · Reviewer_xk2B · 2023-11-01

**Soundness:** 3 good
**Presentation:** 2 fair
**Contribution:** 2 fair
**Rating:** 5
**Confidence:** 4

**Summary:**

The paper proposes KEAI, a framework for domain generalization (DG) using attribute representations and kernel learning. It extracts meaningful attribute features from source domains and uses kernel methods to cluster them into distinct bases with clear decision boundaries. At test time, KEAI identifies which learned attributes are present in the target domain and selectively utilizes them for prediction. The main contributions are 1) an attribute-based representation for DG capturing semantic concepts, 2) a kernel learning approach for robust attribute identification across domains, and 3) achieving strong empirical results compared to prior DG methods. KEAI provides an innovative way to leverage source knowledge when generalizing to new target domains. Experiments validate its effectiveness for domain generalization over the listed baselines.

**Strengths:**

1.	The paper is dedicated to the utilization of target domain-specific information to improve the domain generalization ability of the model, and the proposed method is able to better determine whether a piece of information is present in the target domain or not, compared with the previous methods.
2.	Experiments across standard DG benchmarks demonstrate improvements over the listed baselines.
3.	The proposed techniques are technically sound, with proper formalizations.
4.	The paper is well organized, and the architecture and algorithm descriptions are sufficient.

**Weaknesses:**

1.	The motivation for this paper is not detailed in the introduction. Detailed examples of why and how previous methods have failed to utilize target domain-specific information are needed.
2.	Contributions and novelties of this paper compared to previous studies are not explicitly presented in the paper. The novelty of this paper is limited since the kernel-based methods have been widely studied.
3.	There is no comparison of experimental results with the latest state-of-the-art methods. Important references published after 2021 were not investigated.
4.	Experiments are not sufficient to support claimed benefits, such as determining whether an attribute is present in the target domain.

**Questions:**

1.	What is the main contribution of this paper relative to previous methods, just relying on determining whether an attribute appears in the target domain does not prove that this work has a breakthrough, please clearly describe the contribution and novelty of this paper.
2.	The approach in this paper first performs attribute representation learning and then utilizes a kernel-based approach to categorize the attributes. So, how does the method determine whether an attribute is present in the target domain or not since the training process does not include a structure to process the attribute's origins?
3.	Why are there no up-to-date state-of-the-art methods for comparison, the methods reported in the paper were published in 2021 and before. Additional results with the latest methods are needed to demonstrate the validity of the proposed methods.
4.	The paper does not report the experimental results of the proposed method in identifying whether an attribute appears in the target domain or not. Therefore, the motivation and claimed effectiveness of this method is difficult to prove.
5.	How does the method in this paper perform in a multi-source domain setting? The experimental results in a simple single-source domain setting are not sufficient to prove the superiority of the approach.
6.	Why did the paper not investigate the literature published after 2021 such as [1] and [2].
[1] Pcl: Proxy-based contrastive learning for domain generalization, CVPR2022.
[2] Style neophile: Constantly seeking novel styles for domain generalization, CVPR2022.

---

> ### Author Response · Authors · 2023-11-20
> **Response to Reviewer xk2B (1/2)**
>
> We sincerely appreciate your thorough and insightful feedback. Below, we outline our efforts to address the valuable points you have raised.
>
> &nbsp;
>
> **Contributions and novelties of this paper compared to previous studies are not explicitly presented in the paper. The novelty of this paper is limited since the kernel-based methods have been widely studied.**
>
> *Reply:*
>
> We have expanded the related work section to include additional kernel-based methods, highlighted in blue, and provided a discussion contrasting our approach with these previous methods as follows:
>
> *Different from these existing methods, our proposed approach also falls under kernel-based methods but takes a distinct route. Unlike traditional techniques that utilize the kernel as a tool for measuring divergence or similarity and rely on the kernel trick to avoid explicit computation of the feature map, our method takes a direct approach. We compute the feature map through a random finite-dimensional feature map. This enables us to leverage a clustering-based perspective of kernel methods, which is instrumental in characterizing and refining representations for effective domain generalization.*
>
> We would like to further clarify the key aspects of our contributions as detailed below:
>
> - Attribute-based representations are often expected to be easily interpretable and meaningful to the users of the model. Typically, learning and evaluating such models necessitates specialized domain knowledge. In our research, we reduce the need for such in-depth expertise by aligning our work with a specific downstream task: classification, as outlined in Definition 1.
>
> - We have developed a thorough framework for attribute identification, framing it as a kernel-based optimization problem. In particular, we utilize the clustering perspective of kernel methods for attribute identification. To the best of our knowledge, no prior studies have applied the cluster-induced viewpoint of kernel methods at the attribute level for addressing domain generalization. Our work is pioneering in exploring this novel approach.
>
> - Beyond just identifying bases of attributes, our sophisticated formulation also promotes the distinctness of attribute clusters (**Remark 1**). This approach effectively enhances the discriminative power of the attributes we learn, contributing to the overall robustness and efficacy of our model.
>
> &nbsp;
>
> **The approach in this paper first performs attribute representation learning and then utilizes a kernel-based approach to categorize the attributes. So, how does the method determine whether an attribute is present in the target domain or not since the training process does not include a structure to process the attribute's origins?**
>
> *Reply:*
>
> We formulate the attribute detection in the target domain as an outlier detection problem.
> We start with cluster induced representation of kernel method.
> The hyperplane on kernel space when projecting back into the original attribute space, resulting in a series of contours that envelop the target attribute. These contours serve as cluster boundaries, with points within each distinct contour being assigned to the same cluster.  Additionally, it is found that these contours effectively outline the support of the underlying probability distribution, essentially highlighting high-density regions in the distribution of target attribute. This characteristic is particularly valuable for outlier detection.
>
> During the training process, as we divide the latent space into $K$ groups, all attributes within the same group constitute a basis. To learn the cluster boundaries (hyperplane) $\Gamma_k$ for a basis, we treat attributes from other bases as negatives.
>
> Referring Section 3.4:
>
> *Specifically, the max-margin hyperplane $\Gamma_{k}$ is built in such way that (i) it separates push-forward kernel-features of positive attributes $C\_{+k}=\\{\phi\_{\sigma}(a)\mid a\in\mathcal{A}\_{k}\\}$ and the negative attributes $C\_{-k}=\\{\phi\_{\sigma}(a)\mid a\in\mathcal{A}\_{i\neq k}\\}$ and (ii) the margin w.r.t the hyperplane $\Gamma_{k}$ defined as the closest distance from negative in $C_{-k}$ to this hyperplane
> is maximized.*
>
> It is important to note that our hyperplane in the kernel feature space is integrated with the encoder $g$. Hence, optimizing hyperlanes on kernel space also updates representation of attributes. This integration allows for a more compact refinement of attribute clusters from the basis $A_k$, while also efficiently distinguishing them from attributes derived from other bases.
>
> Upon completion of training, we obtain a set of hyperplanes $\\{\Gamma_k\\}_1^K$, which function as tools for outlier detection. These hyperplanes help in determining whether an attribute in the target domain was learned in the source domain.

---

> ### Author Response · Authors · 2023-11-20
>
> **The paper does not report the experimental results of the proposed method in identifying whether an attribute appears in the target domain or not. Therefore, the motivation and claimed effectiveness of this method is difficult to prove.**
>
> *Reply:*
>
> As kernel-models (hyperplanes) $\\{\Gamma_k\\}_1^K$ characterize high-density regions in the distribution of target attribute,  our model identify whether an attribute in the target domain was learned in the source domain based on set of kernel-model $\\{\Gamma_k\\}_1^K$ in the sense that new attribute in target domain is lied on high-density regions in the distribution of learned attributes from source domain or not.  However, we lack ground-truth labels for qualitative assessment.
>
> To validate this, we use t-SNE for visualizing the distribution of learned attributes, as shown in Figure 4. Our analysis focuses on:
>
> - *Photo* domain which contains rich information to predict the label in comparison with three other domains as the target domain. Hence,It is assumed that the target domain in this case may draw upon information from all source domains.
>
>     From (Figure.4. Right), we observe that the *photo* domain effectively utilizes attributes from all three source domains, as most of its attributes are selected.
>
> - *Sketch* domain which consists solely of colorless images. This choice is intentional, as the *Sketch* domain is markedly different from the others, leading to a more pronounced divergence between source and target.
>
>     In comparison to the *Photo* domain, the *Sketch* domain (as shown in Figure 4, Right) exhibits a smaller number of selected attributes, reflecting its unique characteristics and distinctiveness from other domains. urthermore, it is noticeable that the selected attributes (marked in Yellow) fall within the region of the source domains, while the deselected attributes (indicated in Black) are positioned outside the source domains' region.
>
>
> We have clarified it in Section 4.3.1 (highlighted in blue)
>
> &nbsp;
>
> **How does the method in this paper perform in a multi-source domain setting? The experimental results in a simple single-source domain setting are not sufficient to prove the superiority of the approach.**
>
> *Reply:*
>
> We showcase the performance of our method in multi-source domain settings (using the well-known DomainBed benchmark) in Table 1, and in a straightforward single-source domain scenario in Table 2.
>
> &nbsp;
>
> **Why are there no up-to-date state-of-the-art methods for comparison, the methods reported in the paper were published in 2021 and before. Additional results with the latest methods are needed to demonstrate the validity of the proposed methods.**
>
> *Reply:*
>
> In this work, our primary interest lies in exploring the properties of \textit{kernel-based attributes} and their advantages, rather than pursuing state-of-the-art performance.
>
> Recent approaches, such as [1, 2, 3, 4], frequently adopt an ensemble strategy, particularly one based on SWAD [1]. This approach complicates the evaluation of the effectiveness of their latent representations. As a result, our research primarily focuses on methods that rely on a singular model for evaluation. Nonetheless, to provide a comprehensive view, we have broadened our related works section to include these methodologies. Additionally, we have conducted extra experiments where our KEAI model is trained using an ensemble strategy similar to SWAD, as shown in the following table.
>
> These experiments reveal that KEAI not only performs competitively when compared to state-of-the-art (SOTA) methods but also shows notable effectiveness in domains with limited information, such as the Cartoon and Sketch domains.
>
> &nbsp;
>
> ***Table 1: Classification Accuracy on PACS with ResNet18***
>
> | Algorithm  | Photo | Art-painting | Cartoon | Sketch | Average |
> | :---- | :----: | :----: | :----: | :----: |----: |
> |SWAD [a] | 89.3 +/-0.2 | 83.4 +/-0.6 | 97.3 +/-0.3 | 82.5 +/-0.5 | 88.1|
> |DNA [b] | 89.8 +/-0.2 | 83.4 +/-0.4 | 97.7 +/-0.1 | 82.6 +/-0.2 | 88.4|
> |PCL [1]| **90.2 +/-0.0** | 83.9 +/-0.0 | **98.1 +/-0.0** | 82.6 +/-0.0 | **88.7**|
> |VNE [c] |90.1 +/-0.0 | 83.8 +/-0.0  | 97.5 +/-0.0 |  81.8 +/-0.0 |  88.3|
> |KEAI | 89.2+/-0.3| **84.2+/-0.7**| 97.3+/-0.3 | **83.1+/-0.7**| 88.5|
>
> [1] Pcl: Proxy-based contrastive learning for domain generalization, CVPR2022.
>
> [a] Junbum, Cha et al. (2021). “SWAD: domain generalization by seeking flat minima.” In: NeurIPS.
>
> [b] Xu, Chu et al. (2022). “DNA: domain generalization with diversified neural averaging.” In: ICML.
>
> [c] Kim, Jaeill, et al. "VNE: An Effective Method for Improving Deep Representation by Manipulating Eigenvalue Distribution." C. PRV 2023.

---

> > ### Author Response · Authors · 2023-11-21
> >
> > **Why did the paper not investigate the literature published after 2021 such as [1] and [2]. [1] Pcl: Proxy-based contrastive learning for domain generalization, CVPR2022. [2] Style neophile: Constantly seeking novel styles for domain generalization, CVPR2022.**
> >
> > *Reply:*
> >
> > We have discussed the papers the reviewer suggested in the revised version. Note that those works focused on domain-invariant representation learning approaches based on unsupervised learning [1] and augmentation [2] which differ from our approach which aims to capture diverge attributes from source domains. Additionally, we have expanded the related work section to encompass more recent kernel-based methods.

---

### Meta-Review · Area_Chair_oNRw · 2023-12-07

**Metareview:**

Thanks for your submission to ICLR.

This paper presents an approach to domain generalization using attribute representations and kernel learning.  The reviewers noted several drawbacks/concerns, including: limited novelty, missing experiments to state-of-the-art, missing references, and unconvincing experiments.  All four reviewers were in agreement that the paper is not ready for publication, even after the rebuttal period.

**Justification For Why Not Higher Score:**

All four reviewers are in agreement here that the paper is not ready for publication.

**Justification For Why Not Lower Score:**

N/A

---

### Decision · Program_Chairs · 2024-01-16

Reject